# TWO STEP DIFFUSION: FAST SAMPLING AND RELIABLE PREDICTION OF 3D KELLER-SEGEL CHEMOTAXIS SYSTEMS IN FLUID FLOWS

## ABSTRACT

In this work, we study fast and reliable generative transport for the 3D Keller-Segel in different fluid flows, where the goal is to map initial particles $x_0$ to terminal states $x_1$ for a range of physical parameters $\sigma$. While the quadratic Wasserstein distance $W_2$ serves as a better metric for differences between distributions, optimizing $W_2$ *directly* is unstable and computationally expensive in high dimensions. We propose a two-stage pipeline that retains one-step efficiency while reinstating an explicit $W_2$ objective where it is tractable. In **Stage I**, a MeanFlow-style regressor trained via the MeanFlow identity yields a deterministic, 1-NFE global transport that moves particles close to their terminal states without simulating forward diffusions. In **Stage II**, we freeze this initializer and train a near-identity corrector (*Deep Particle, DP*) that *directly* minimizes a mini-batch $W_2$ objective using warm-started EMD (Earth Mover's Distance)/OT (Optimal Transport) couplings computed on the MeanFlow outputs. Crucially, after the one-step transport (from Stage I) has concentrated mass on the correct support, the induced geometry stabilizes high-dimensional $W_2$ optimization. We validate our construction on the 3D Keller Segel setting with different flows. We find that our method consistently reduces the empirical $W_2$ relative to one-step flows. Moreover, the two-stage refinement **reduces MeanFlow's generalization error** across $\sigma$, demonstrating improved robustness to parameter shift, which is evidenced by consistently lower $W_2$ over the $\sigma$ sweep. At the same time, the approach preserves the properties of 1-NFE and deterministic sampling, and yields clear qualitative gains in anisotropy and mass placement, as supported by our simulations and $W_2$-vs-$\sigma$ curves.

**Keywords:** 3D Keller-Segel chemotaxis systems; Two-Step Diffusion; Optimal Transport (OT); Wasserstein-2 distance ($W_2$); GPU-friendly mini-batch OT training.

## 1 INTRODUCTION

Learning parameterized particle transports that faithfully match simulator outputs while remaining fast at inference is a core challenge in flow-based generative modeling Lipman et al. (2022); Liu et al. (2022) and scientific machine learning. Keller Segel systems were introduced by Keller and Segel to describe the aggregation of the slime mold Dictyostelium discoideum in response to an attractive chemical signal Keller & Segel (1970), which is hard to simulate the particle distribution with fluid flows. For the setting considered here, the goal is to map initial particles $x_0$ to terminal states $x_1$ across a range of physical parameters $\sigma$, with an objective that reflects the geometry of probability measures. The quadratic Wasserstein distance ($W_2$) is an attractive choice for this purpose Peyré et al. (2019), yet *directly* optimizing $W_2$ in high dimensions is notoriously difficult Genevay et al. (2016): computing couplings is expensive, gradients are noisy Genevay et al. (2019), and naive mini-batch surrogates can be unstable Nguyen et al. (2022).
Recent few-/one-step flow methods make impressive progress on fast, deterministic sampling by regressing velocity fields from paired endpoints without simulating forward diffusions Salimans & Ho (2022); Geng et al. (2023). These approaches scale well to high-dimensional data because they bypass explicit optimal-transport (OT) solves and the current state of the art method is the Meanflow Geng et al. (2025) method proposed in 2025. However, *MeanFlow does not directly*

*target the Wasserstein distance*. As a result, although its one-step predictions often land near the correct support, residual misalignment in mass placement and anisotropy may persist—particularly in stiff, multi-scale regimes.

To tackle this, we propose a **two-stage** pipeline that preserves the speed and stability of one-step flows while restoring an explicit $W_2$ training signal where it matters most. In **Stage I** (Algorithm 1), we train a MeanFlow-style regressor using the meanflow identity to obtain a *deterministic, 1-NFE* global transport that moves particles close to terminal states across physical parameters. In **Stage II** (Algorithm 1), we freeze this initializer and train a near-identity corrector (*Deep Particle, DP*) whose loss *directly minimizes a mini-batch approximation to $W_2$* via a warm-started coupling. The key observation is that after the fast one-step transport has concentrated mass on (or near) the correct support, the induced geometry makes $W_2$ easy to optimize: couplings are closer to permutation-like, costs are locally well-conditioned, and mini-batch OT becomes stable and GPU-friendly. Thus, instead of attacking a hard high-dimensional $W_2$ optimization from scratch, we first simplify the geometry with a one-step flow, then solve a much easier, local $W_2$ refinement. This two-step design *avoids heavy global OT computation* while giving us a principled, geometry-aware objective.

**Key Contributions.**

1. A **two-stage** KS-3D-Laminar solver: a **1-NFE MeanFlow** initializer for global transport followed by a **near-identity DP corrector** that *directly* minimizes a mini-batch $W_2$ objective with warm-started OT couplings.

2. A demonstration that this decomposition *turns an intractable high-dimensional $W_2$ problem into a tractable one*: the first step simplifies geometry; the second step performs local, measure-aware alignment without heavy global OT.

3. Empirical gains across a wide $\sigma$ sweep, especially in the singular perturbation regime and reducing empirical $W_2$ while preserving determinism and low NFE.

## 2 RELATED WORK

### 2.1 DIFFUSION MODEL AND FLOW MATCHING METHOD

Classical *diffusion models* Sohl-Dickstein et al. (2015); Song & Ermon (2019); Ho et al. (2020) learn a score field $\nabla_z \log p_t(z)$ by denoising score matching while simulating (or reversing) a noising SDE Ho et al. (2020); samples are generated by integrating a reverse *stochastic* process, typically requiring many function evaluations (NFEs). Subsequent work (e.g., consistency( Luo et al. (2023); Song et al. (2023))/ODE distillation Zhou et al. (2024); Yin et al. (2024)) compresses sampling to a few steps but often relies on auxiliary training stages. In contrast, *Flow Matching (FM)* Lipman et al. (2022); Albergo & Vanden-Eijnden (2022)) learns a *deterministic* velocity field $v_\theta(z, t)$ via supervised regression on analytically known *conditional velocities* along simple paths between data $x_0$ and prior $x_1$ (e.g., a linear interpolation). The key identity is that the unknown *marginal* velocity equals the conditional expectation of conditional velocities, enabling stable one-stage training sampling without ever estimating scores or simulating an SDE Lipman et al. (2022). The target and prior for the flow matching are $x_0 \sim \pi_0, x_1 \sim \pi_1$ (typically $\mathcal{N}(0, I)$), the generative ODE is the following:

$$\dot{z}_t = v_\theta(z_t, t), \qquad t \in [0, 1] \tag{1}$$

with a linear conditional path:

$$z_t := (1 - t)x_0 + tx_1, \qquad t \in [0, 1] \tag{2}$$

and its conditional velocity is just the time derivative:

$$v^*(z_t, t \mid x_0, x_1) = \frac{d}{dt} z_t = x_1 - x_0 \tag{3}$$

and the marginal ground-truth velocity is the conditional expectation:

$$v^*(z, t) = \mathbb{E}\left[v^*(z_t, t \mid x_0, x_1) \mid z_t = z\right] \tag{4}$$

which means that we can train with conditional targets and still learn the correct marginal field.

## 2.2 ONE-STEP FLOW-MATCHING MODEL VIA MEANFLOW

Standard *Flow Lipman et al. (2022)* learns the marginal velocity field $v^*(z, t)$, which typically induces *curved* Eulerian trajectories; when combined with a *low-NFE* numerical solver, the ODE integration can be *inaccurate*, leading to endpoint bias and distributional mismatch (often exacerbated by guidance). By contrast, **MeanFlow Geng et al. (2025)** targets the *average (endpoint) displacement* directly and leverages an analytical mean–flow identity for supervision, yielding a natural **true 1-NFE** sampler that avoids time integration and is less sensitive to trajectory curvature. For $0 \leq r < t \leq 1$, define

$$u(z_t, r, t) = \frac{1}{t - r} \int_r^t v(z_\tau, \tau) \, d\tau. \tag{5}$$

As $r \to t$, $u(z_t, r, t) \to v(z_t, t)$. Moreover, additivity of the integral implies the *consistency relation Song et al. (2023)*: for any $r < s < t$,

$$(t - r) \, u(z_t, r, t) \;=\; (s - r) \, u(z_s, r, s) \;+\; (t - s) \, u(z_t, s, t). \tag{6}$$

Rewriting (5) as $(t - r) u(z_t, r, t) = \int_r^t v(z_\tau, \tau) \, d\tau$ and differentiating w.r.t. $t$ (with $r$ fixed) gives

$$u(z_t, r, t) \;=\; v(z_t, t) \;-\; (t - r) \frac{d}{dt} u(z_t, r, t), \tag{7}$$

Equations (5) and (7) are equivalent; (7) is the *Mean–Flow Identity Geng et al. (2025)*.

Meanflow method offers one NFE and deterministic sampling with simple conditioning and well-conditioned averaged-velocity targets, but can retain residual bias on multi-modal/multi-scale data, is sensitive to time sampling and JVP accuracy, and often benefits from a lightweight corrector (e.g., OT refinement( Tong et al. (2023a)).

## 2.3 WASSERSTEIN AND DISCRETE WASSERSTEIN DISTANCE

The Wasserstein distance provides a geometry on probability measures by quantifying the minimal "effort" required to morph one distribution into another Villani (2021) . For probability measures $\mu$ and $\nu$ on a metric space $(Y, \mathrm{dist})$ and $p = 2$, the quadratic Wasserstein distance is

$$W_2(\mu, \nu) = \left( \inf_{\gamma \in \Gamma(\mu, \nu)} \int_{Y \times Y} \mathrm{dist}(y', y)^2 \, d\gamma(y', y) \right)^{1/2}, \tag{8}$$

where $\Gamma(\mu, \nu)$ is the set of couplings (joint measures) with marginals $\mu$ and $\nu$. When a map $f : X \to Y$ pushes $\mu$ forward to $f_* \mu$, the distance between the transformed source and target can be written as

$$W_2(f_* \mu, \nu) = \left( \inf_{\gamma \in \Gamma(\mu, \nu)} \int_{X \times Y} \mathrm{dist}(f(x), y)^2 \, d\gamma(x, y) \right)^{1/2}. \tag{9}$$

For computation Peyré et al. (2019), one typically replaces $\mu$ and $\nu$ by empirical measures with $N$ samples, $\mu = \frac{1}{N} \sum_{i=1}^N \delta_{x_i}$, $\nu = \frac{1}{N} \sum_{j=1}^N \delta_{y_j}$. In this discrete setting, a coupling $\gamma$ becomes a nonnegative $N \times N$ matrix $(\gamma_{ij})$ with unit row/column sums (a doubly stochastic matrix Sinkhorn (1964)),

$$\gamma_{ij} \geq 0, \qquad \sum_{i=1}^N \gamma_{ij} = 1, \quad \sum_{j=1}^N \gamma_{ij} = 1, \tag{10}$$

so the discrete quadratic Wasserstein objective reduces to the linear program

$$\widehat{W}_2(f) = \left( \inf_{\gamma \in \Gamma^N} \frac{1}{N} \sum_{i=1}^N \sum_{j=1}^N \mathrm{dist}(f(x_i), y_j)^2 \, \gamma_{ij} \right)^{1/2}. \tag{11}$$

Here, $\gamma_{ij}$ represents the proportion of mass moved from $f(x_i)$ to $y_j$; the optimal value is the minimum transport "effort" needed to align $f_* \mu$ with $\nu$. In practice, this discrete formulation underlies modern optimal transport solvers and learning pipelines, often estimated on mini-batches and accelerated via entropic regularization and Sinkhorn iterations Altschuler et al. (2017), which trade exactness for speed while preserving the geometric bias of optimal transport.

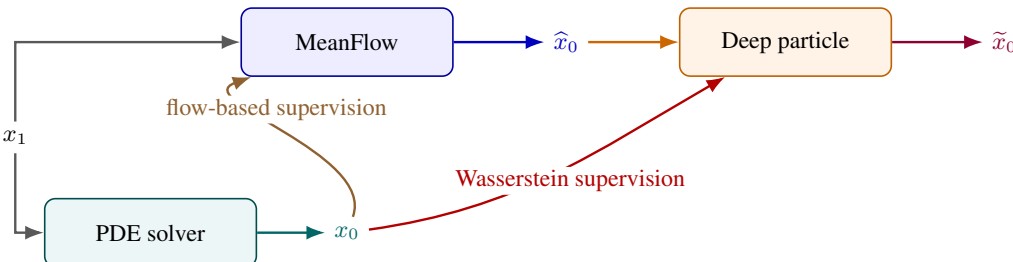

Figure 1: Two-step pipeline. MeanFlow produces an initial estimator $\widehat{x}_0$ with flow-based supervision, the PDE solver provides $x_0$, and Deep particle refines to $\widetilde{x}_0$ via Wasserstein Distance supervision.

## 3 METHODOLOGY

### 3.1 PART I: MEANFLOW AS A ONE-STEP GENERATIVE PROCESS

Meanflow Geng et al. (2025), in the Flow–Matching view, is learning a velocity field $v(z,t)$ which transports the prior at $t=1$ to the data at $t=0$. Along the interpolation $z_t = (1-t)\,x_0 + t\,x_1$. The endpoint relation is

$$x_0 \;=\; x_1 - \int_0^1 v(z_t, t)\, dt. \tag{12}$$

Define the *average (mean) velocity* between $(r,t) = (0,1)$ by

$$u(x_1, 0, 1) \;=\; \int_0^1 v(z_t, t)\, dt. \tag{13}$$

Combining (12)–(13) yields the one–step sampling map

$$x_0 \;=\; x_1 - u(x_1, 0, 1) = T(x_1). \tag{14}$$

In practice we train a network $u_\theta$ to approximate $u(\cdot, 0, 1)$ and use the single forward pass $\hat{x}_0 = \varepsilon - u_\theta(\varepsilon, 0, 1, \boldsymbol{\sigma})$ for sampling.

Although MeanFlow is less sensitive to trajectory curvature Geng et al. (2025) than direct few-step ODE integration (it predicts the path integral in (13) directly), it remains within the Flow–Matching paradigm and exhibits several sources of error when used alone: (i) **Curvature bias:** the marginal field $v(\cdot, t)$ typically induces *curved* characteristics; learning a single displacement $u(\cdot, 0, 1)$ tends to capture coarse global transport but smooths out fine structures. (ii) **Path mismatch:** the linear training path $z_t$ need not coincide with the true characteristics of the marginal flow; the learned average may therefore underfit localized details. These effects are amplified in **physical science** settings, where data are low-dimensional but the tolerance on small discrepancies is high; recovering the distributional detail may demand disproportionately long training or larger models.

### 3.2 PART II: WASSERSTEIN REFINEMENT (BRINGING DISTRIBUTIONS CLOSER, THEN ALIGNING)

To correct the residual bias from the one–step map (14), we refine the MF outputs by *explicitly* minimizing a Wasserstein distance between the generated and reference distributions. $W_2$ provides a geometry–aware metric that measures distributional misalignment even when supports do not overlap, making it well-suited for precise alignment in low dimensions. However, its computation is expensive: (i) constructing the pairwise cost matrix is $\mathcal{O}(N^2)$ in both time and memory for a batch of size $N$; (ii) Exact solvers scale superlinearly and entropic or accelerated variants (e.g., Sinkhorn) require many iterations when the two distributions are *far apart* or when small regularization is needed for high accuracy. Thus, a naive end–to–end $W_2$ minimization from a poor initialization can be prohibitively slow.

We therefore adopt a *coarse–to–fine* pipeline which is clearly shown in the figure 1:

1. **Stage I (MeanFlow).** Use the one-step generator (14) to move samples quickly along the global transport, bringing the model distribution close to the target with **1–NFE** per sample.

2. **Stage II (Wasserstein refinement).** Starting from these near–aligned particles, minimize a mini–batch approximation of $W_2$ to correct remaining local misalignments. Because the two distributions are now close, the OT coupling converges rapidly and the per-iteration cost is dominated by a manageable $\mathcal{O}(N^2)$ cost–matrix build.

For stability and precision, the refinement map is parameterized as a *near–identity* ResNet He et al. (2016), which is well conditioned for small corrections and empirically attains higher accuracy than unrestricted parametrizations. To improve the computation efficiency, we employed the deep particle method( Wang et al. (2024); Zhang et al. (2025)), which use minibatch OT interior points solver to approximate the Wasserstein distance. Given mini-batches $\{x_i^{\mathrm{mf}}\}$ and $\{x_j^{\mathrm{ref}}\}$, we form $C_{ij} = \|f_\phi(x_i^{\mathrm{mf}}, \sigma) - x_j^{\mathrm{ref}}\|_2^2$ and update a doubly-stochastic coupling $\gamma$ using a standard mini-batch OT routine ; the training objective is $\mathcal{L}_{\mathrm{refine}} = \langle C, \gamma \rangle + \lambda_{\mathrm{res}} \sum_i \|f_\phi(x_i^{\mathrm{mf}}, \ell) - x_i^{\mathrm{mf}}\|_2^2$, which preserves the near-identity structure while aligning residual discrepancies.

This two–stage design avoids (i) **the heavy training burden** MeanFlow would incur to recover distributional details on its own, and (ii) **the large computational overhead** of optimizing Wasserstein distance when the two distributions start far apart, while markedly improving final accuracy. The two-stage procedure is summarized in Algorithm 1.

With this Two-Stage Design, we can enjoy from: **(i) Fast global transport.** MF offers 1-NFE sampling and serves as a strong initializer that moves particles close to terminal states, drastically reducing refinement burden. **(ii) Stable training signals.** The JVP-based residual with adaptive weighting provides a well-conditioned objective that is robust to outliers and multi-scale dynamics. **(iii) Geometry-aware local alignment.** Mini-batch $W_2$ couplings refines MF outputs, where barycentric pulls derived from $\gamma$ correct small residual biases while preserving near-identity structure, largely shortening the gap between the distributions.

To be noted, MeanFlow cannot be used as a second stage after Deep Particle (DP): MeanFlow is restricted to mappings from a Gaussian base to the target, whereas DP transports between arbitrary distributions.

### 3.3 NETWORK ARCHITECTURE

Following Parts I–II, we instantiate two modules consistent with the notation above. The transport predictor $u_\theta : \mathbb{R}^d \times [0,1]^2 \times \mathbb{R} \to \mathbb{R}^d$ takes $z$ concatenated with sinusoidal embeddings of $(r, t, \sigma)$, where $\sigma$ is the embedded physical parameter, and is implemented as a compact MLP (width 64, depth 5, SELU, light skip connections) with a small-gain linear head; during training we evaluate $(u, \dot{u})$ via a single forward-mode JVP along the direction $(v, 0, 1)$ to obtain $\dot{u} = \partial_t u(z, r, t) + (\nabla_z u(z, r, t)) v$, as required by (7), while sampling uses the endpoint map in (14) with $u_\theta(\cdot, 0, 1, \sigma)$. The refinement map $f_\phi : \mathbb{R}^d \times \mathbb{R} \to \mathbb{R}^d$ is parameterized as a near-identity ResNet, $f_\phi(x, \ell) = x + \alpha\, h_\phi([x, \ell])$, depth 4, width 64, SiLU activations and LayerNorm, to focus capacity on small corrections introduced in Part II.

## 4 EXPERIMENTS

### 4.1 3D KELLER SEGEL SYSTEM WITH 2D LAMINAR FLOW

In this experiment, we simulate particle density evolution under chemotaxis and a steady 3D laminar flow. In the singular limit, the particle density $\rho$ evolves according to:

$$\rho_t = \mu\, \Delta\rho + \chi\, \nabla \cdot \big(\rho\, \nabla(K * \rho)\big) \tag{15}$$

where $\rho$ is the density of the bacteria ,$\mu, \chi, \epsilon, k$ are non-negative constants,$K(x) = -\frac{1}{8\pi}\|x\|^{-1}$ is the Green's function of the Laplacian in three dimensions. The background velocity is chosen as a divergence–free 2D Laminar flow:

---

**Algorithm 1** Two-Stage Training: MeanFlow (Stage I) then Wasserstein DP Refinement (Stage II)

---

**Require:** Paired endpoints $\{(x_0^{(i)}, x_1^{(i)}, \sigma^{(i)})\}$;
 1:      MF network $u_\theta(z, t, r, \boldsymbol{\sigma})$ taking $(z, t, r, \boldsymbol{\sigma})$;
 2:      DP map $f_\phi(x, \boldsymbol{\sigma}) = x + \alpha\, h_\phi([x, \boldsymbol{\sigma}])$ with zero-initialized last layer;
 3:      OT solver (e.g., EMD) for mini-batch coupling $\gamma$;
 4:      MF batch size $B_{\text{mf}}$, MF steps $T_{\text{mf}}$;
 5:      DP batch size $B_{\text{dp}}$, DP iterations $T_{\text{dp}}$;
 6:      DP batch/plan refresh periods $S_{\text{batch}}, S_\gamma$;
 7:      near-identity scale $\alpha$, residual weight $\lambda_{\text{res}} > 0$.

 8: **Stage I: Train MeanFlow by MeanFlow Identity (JVP regression)**
 9: Build endpoint table by concatenation: $X^{\text{pairs}} = \{(x_0^{(i)}, x_1^{(i)}, \sigma^{(i)})\}_i$
10: **for** $k = 1$ to $T_{\text{mf}}$ **do**
11:      Sample a dictionary index (if applicable) and a mini-batch of size $B_{\text{mf}}$:
         $(x_0, x_1, \sigma) \sim X^{\text{pairs}}$, with $\ell = \boldsymbol{\sigma}$
12:      Sample times $(t, r)$ with $t \geq r$ (e.g., logit-normal); set $r=t$ for half the batch
13:      Set linear path :   $z \leftarrow (1-t)\, x_0 + t\, x_1$;    conditional velocity $v \leftarrow x_1 - x_0$
14:      Evaluate JVP to obtain the *total* time derivative along the path:
$$(u, \dot{u}) \leftarrow \text{JVP}\Big((z, t, r) \mapsto u_\theta(z, t, r, \ell),\ (z, t, r);\ (v, 1, 0)\Big)$$
15:      Form the Mean–Flow target via identity $u = v - (t-r)\frac{d}{dt}u$:
         $u_{\text{tgt}} \leftarrow v - (t-r)\, \dot{u}$                                    (no grad through $u_{\text{tgt}}$)
16:      Compute loss (optionally with robust/adaptive weighting):
         $\mathcal{L}_{\text{MF}} \leftarrow \frac{1}{B_{\text{mf}}} \sum \big\| u - \text{sg}[u_{\text{tgt}}] \big\|_2^2$
17:      Update $\theta \leftarrow \theta - \eta_\theta \nabla_\theta \mathcal{L}_{\text{MF}}$
18: **end for**
19: **Freeze** $\theta$

20: **Stage II(a): Build DP training pairs using MF one-step outputs**
21: **for** each conditioning value $\sigma$ in the training grid **do**
22:      Draw $N_{\text{dp}}$ fresh prior samples $\varepsilon \sim \pi_1(\sigma)$
23:      One-step MF generation (note the *subtraction*, consistent with the training path):
         $x_{\text{mf}} \leftarrow \varepsilon - u_\theta(\varepsilon,\, t=1,\, r=0,\, \boldsymbol{\sigma})$
24:      Draw references $x_{\text{ref}} \sim \mathcal{S}(\sigma)$                  ▷ ground-truth / simulator targets
25:      Store paired mini-batches $\big([x_{\text{mf}}, \boldsymbol{\sigma}],\ x_{\text{ref}}\big)$
26: **end for**

27: **Stage II(b): Train the near-identity DP corrector with mini-batch OT**
28: Initialize $\phi$ (last layer zeros); set iteration counter $t \leftarrow 0$
29: Initialize uniform coupling per mini-batch: when batch size is $B_{\text{dp}}$,
         $\gamma \leftarrow \frac{1}{B_{\text{dp}}} \mathbf{1}\mathbf{1}^\top \in \Gamma_{B_{\text{dp}}}$ (row/col sums $= \frac{1}{B_{\text{dp}}}$)
30: **for** $t = 1$ to $T_{\text{dp}}$ **do**
31:      **if** $t \bmod S_{\text{batch}} = 0$ **then**
32:          Sample fresh DP mini-batch $\{x_{\text{mf}}, x_{\text{ref}}, \sigma\}$ of size $B_{\text{dp}}$
33:      **end if**
34:      Compute refined points $x_\phi \leftarrow f_\phi(x_{\text{mf}}, \boldsymbol{\sigma})$
35:      **if** $t \bmod S_\gamma = 0$ **then**
36:          Build cost $C_{ij} \leftarrow \|x_{\phi,i} - x_{\text{ref},j}\|_2^2$
37:          Update coupling $\gamma \leftarrow \arg\min_{\gamma \in \Gamma_{B_{\text{dp}}}} \langle C, \gamma \rangle$         ▷ mini-batch OT, e.g., EMD
38:      **end if**
39:      OT loss with residual regularization:
         $\mathcal{L}_{\text{DP}} \leftarrow \langle C, \gamma \rangle\ +\ \lambda_{\text{res}} \frac{1}{B_{\text{dp}}} \sum_i \|x_{\phi,i} - x_{\text{mf},i}\|_2^2$
40:      Update $\phi \leftarrow \phi - \eta_\phi \nabla_\phi \mathcal{L}_{\text{DP}}$
41: **end for**
42: **Output:** trained $\theta$ (MeanFlow) and $\phi$ (DP corrector)

---

$$\mathbf{v}(x, y, z) \;=\; \sigma\left(e^{-y^2 - z^2}, \, 0, \, 0\right)^T, \tag{16}$$

Consequently, $\sigma$ scales the flow strength at X-axis. The ground–truth particle dynamics are generated through an interacting particle approximation of (15), namely

$$dX_j = -\frac{\chi}{J} \sum_{i \neq j} \nabla K_\delta(|X_i - X_j|) \, dt + v(X_j) \, dt + \sqrt{2\mu} \, dW_j, \tag{17}$$

where $K_\delta(z) = K(z)\frac{|z|^2}{|z|^2 + \delta^2}$, employing the parameter $\delta$ to regularize the kernel.

### 4.1.1 PROBLEM SETUP AND DATA GENERATION

We consider the 3D–KS–Laminar setting generate training pairs $(x_0, x_1)$ by simulating particle ensembles under different advection amplitudes $\sigma$. Specifically, we take $n_{\text{dict}} = 8$ logarithmically spaced values $\sigma \in 1.5 \cdot 10^{\text{linspace}(1,2,8)}$, and initialize $N_{\text{particles}} = 15{,}000$ samples in $\mathbb{R}^3$ for each choice of $\sigma$. The forward simulation runs up to $T = 0.02$ with step size $\Delta t = 5 \times 10^{-3}$. The initial configuration $x_0$ is shared across all $\sigma$ for alignment, and the evolved states $x_1$ form the target data.

### 4.1.2 EXPERIMENT ANALYSIS

Across $\sigma$, DP strictly lowers the 2-Wasserstein metric $W_2$ relative to MF, with the largest gains in the singular-perturbation regime. In the interpolation range ($\sigma \in \{20, 40, \dots, 140\}$) improvements are modest but consistent (e.g., $0.0201 \to \mathbf{0.0065}$ at $\sigma{=}120$). In contrast, the extrapolation regime shows dramatic improvements: at $\sigma{=}160$, MF's $W_2$ drops from $0.0403$ to $\mathbf{0.0082}$ ($\approx 4.9\times$ smaller); at $\sigma{=}180$, from $0.0832$ to $\mathbf{0.0140}$ ($\approx 5.9\times$); and at $\sigma{=}200$, from $0.1970$ to $\mathbf{0.0214}$ ($\approx 9.2\times$). The right-hand curve in Fig. 2 shows MF's error rising sharply beyond $\sigma = 150$, while DP remains low and flat. This indicates that a one-step MF sampler followed by a lightweight OT-based corrector sustains accuracy even under strong advection, further underscoring the method's generalization capacity. At $\sigma = 160$, an *extrapolation* point outside the training grid of the reference projections in Fig. 3 exhibits the expected anisotropic advection along $x$: mass near $(y, z) \approx (0, 0)$ is transported farther in $x$ than mass at larger $|y|$ or $|z|$, yielding a tapered fan in $(x, y)$ and $(x, z)$ and an approximately isotropic footprint in $(y, z)$. One-step MeanFlow (MF) captures the global advection but remains overly diffuse and $x$–biased. Applying the near-identity DP corrector contracts the distribution toward the reference barycenter and restores the anisotropic taper where fans narrow with steeper central gradients while the $(y, z)$ slice regains circularity, without introducing artifacts. That this behavior holds at $\sigma = 160$ (extrapolation) highlights the strong out-of-distribution generalization ability of our two-step pipeline.

| $\sigma$ | Meanflow | DP refinement |
|---|---|---|
| $20^{(\blacktriangle)}$ | 0.0084 | **0.0047** |
| $40^{(\blacktriangle)}$ | 0.0068 | **0.0046** |
| $60^{(\blacktriangle)}$ | 0.0070 | **0.0049** |
| $80^{(\blacktriangle)}$ | 0.0105 | **0.0056** |
| $100^{(\blacktriangle)}$ | 0.0132 | **0.0059** |
| $120^{(\blacktriangle)}$ | 0.0201 | **0.0065** |
| $140^{(\blacktriangle)}$ | 0.0216 | **0.0073** |
| $160^{(\bullet)}$ | 0.0403 | **0.0082** |
| $180^{(\bullet)}$ | 0.0832 | **0.0140** |
| $200^{(\bullet)}$ | 0.1970 | **0.0214** |

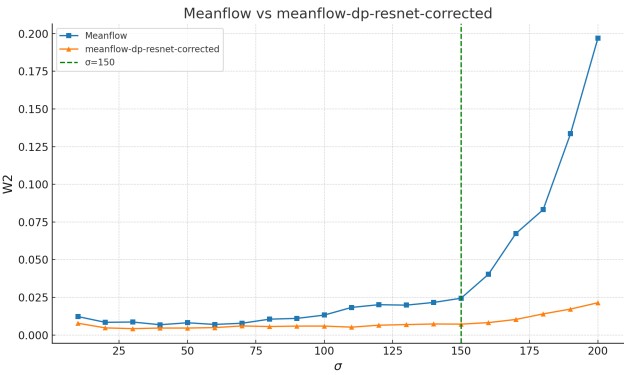

Figure 2: $W_2$ across $\sigma$ (left table) and $W_2$ vs. $\sigma$ (right). Superscripts on $\sigma$ denote usage: $\circ$—*training*, $\blacktriangle$—*interpolation*, $\bullet$—*extrapolation*. DP refinement consistently achieves the lowest cost and shows the largest gains in the singular perturbation regime ($\sigma \gtrsim 150$). MF uses one-shot sampling; DP is the refinement in Algorithm 1.

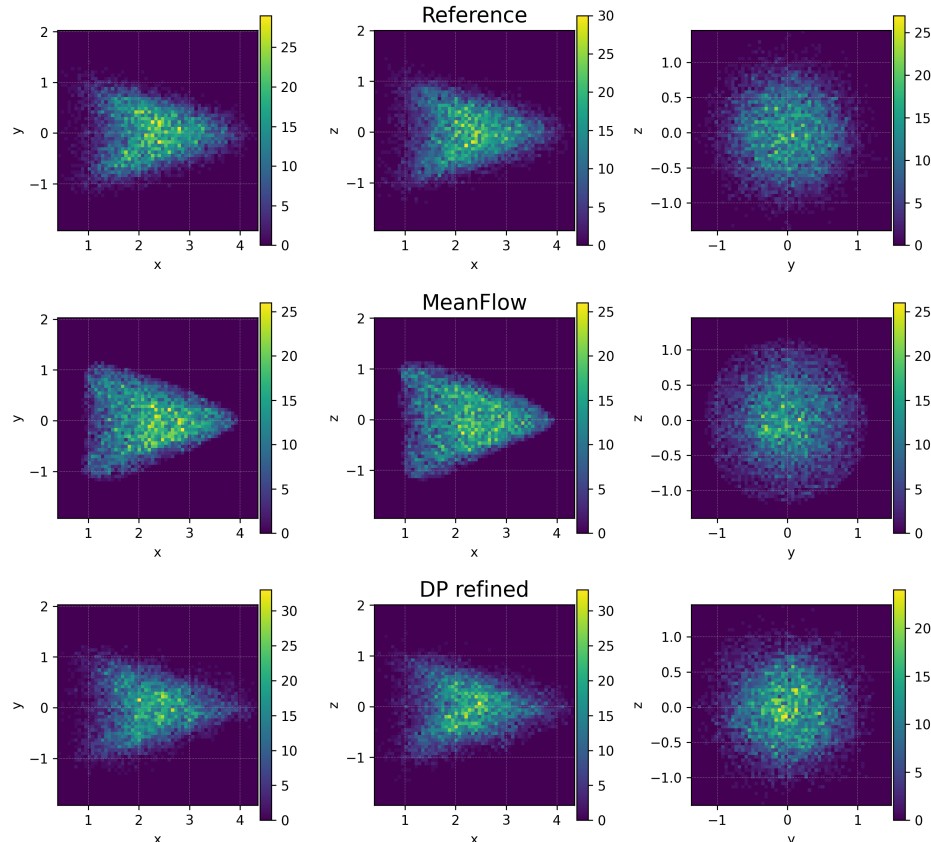

Figure 3: Qualitative comparisons on $(x, y), (x, z), (y, z)$ at $\sigma = 160$ for the 3D Keller Segel system with 2D Laminar flow. (a) Reference solution projected to three coordinate plains. (b) Predicted solution projected to the three coordinate plains by MeanFlow($W_2 = 0.0403$) (c) DP Refinement solution projects to the three coordinate plains ($w_2 = 0.0082$).

### 4.2 3D KELLER SEGEL SYSTEM WITH 3D KOLMOGOROV FLOW

In this experiment, we deal with the 3D keller Segel system (equation 15) with 3D Kolmogorov flow:

$$\mathbf{v}(x, y, z) = \sigma \cdot \big(\sin(2\pi z), \sin(2\pi x), \sin(2\pi y)\big)^T. \qquad (18)$$

The data generation process is almost the same except we use $n_{\text{dict}}=10$ values on an equal-spaced grid $\sigma \in \text{linspace}(10, 100, 10)$ and check with the extrapolation result when $\sigma = 110$.

#### 4.2.1 EXPERIMENT ANALYSIS

At $\sigma = 110$, an extrapolation point beyond the training grid $\sigma \in \{10, \dots, 100\}$,the reference projections in Fig. 4 display the Kolmogorov-flow–induced anisotropy: narrow, high-density ridges aligned with coordinate directions in $(x, y)$ and $(y, z)$, together with a more isotropic footprint in $(x, z)$. One–step MeanFlow captures the gross concentration but remains over-smoothed, attenuating the ridge contrast and slightly biasing the barycenter. The DP corrector (near-identity, mini-batch OT) sharpens the directional structures: ridges become thinner and more pronounced in $(x, y)$ and $(y, z)$ while the $(x, z)$ slice regains the compact, near-circular core. Crucially, this improvement at $\sigma=110$ (extrapolation) indicates that our two-step pipeline generalizes robustly out of distribution while preserving the 1-NFE determinism of MF.

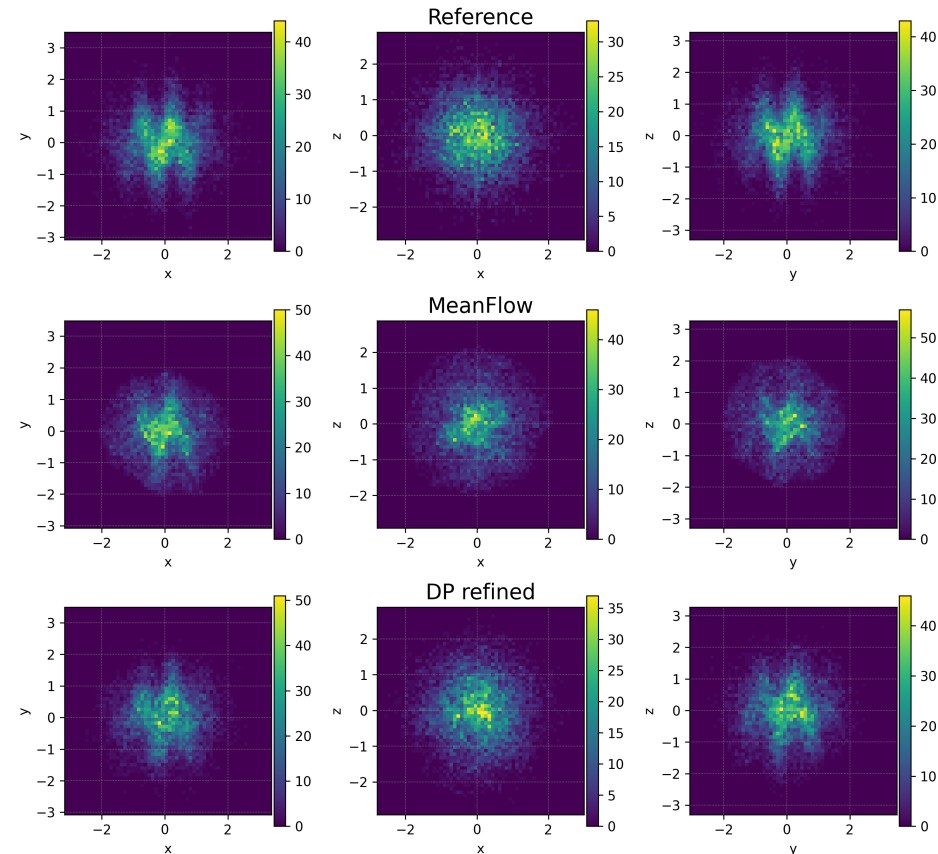

Figure 4: Qualitative comparisons of $(x, y)$, $(x, z)$, $(y, z)$ plains at $\sigma = 110$ for the 3D Keller Segel system with 3D Kolmogorov flow. (a) Reference solution projected to three coordinate plains. (b) Predicted solution projected to the three coordinate plains by MeanFlow (c) DP Refinement solution projects to the three coordinate plains.

## 5 CONCLUSION

We introduced a Two-Step Diffusion framework for fast and reliable generation in Keller–Segel chemotaxis systems with background flows. The method separates coarse global transport from fine local alignment: a MeanFlow-style regressor, trained via the MeanFlow identity, provides a deterministic, 1-NFE initializer that moves particles close to the correct support, and a near-identity DeepParticle (DP) corrector then minimizes a mini-batch Wasserstein-2 objective with exact EMD couplings. This coarse-to-fine design restores an explicit geometry-aware $W_2$ training signal where it is tractable, making high-dimensional OT optimization stable while preserving one-shot sampling.

Across 3D Keller–Segel systems with both laminar and Kolmogorov flows, the two-stage pipeline consistently lowers empirical $W_2$ relative to MeanFlow and sharpens anisotropic structure, with the largest gains in the singular-perturbation and extrapolation regimes. The Appendix further shows that these benefits strengthen in a 4D KS extension, are robust under changes in mini-batch size and coupling refresh rate, and persist when comparing against self-distilled IMM and OT-informed SF2M baselines as well as regularized Sinkhorn solvers. In Kolmogorov–Petrovsky–Piskunov (KPP) front-speed experiments on both 2D and 3D time-dependent flows, the MF+DP warm start yields invariant measures substantially closer to long-horizon Feynman–Kac references and eigen-value estimates that are nearly unbiased from the first generation, highlighting Two-Step Diffusion as a practical recipe for combining fast one-step flows with principled OT-based refinement in sci-entific ML.

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

## 6 APPENDIX

This appendix provides additional experiments and analyses that complement the main text. Section 6.1 extends the Keller–Segel setup to a 4D laminar flow and shows that the advantage of our two-stage design becomes even more pronounced in higher dimensions. Section 6.2 presents an ablation over the DP mini-batch size and coupling refresh rate, demonstrating that our refinement is both efficient and robust to these hyperparameters. Section 6.3 compares Two-Step Diffusion with IMM-style self-distilled one-step baselines and OT-informed SF2M models, while Section 6.4 contrasts exact mini-batch EMD with entropically regularized Sinkhorn solvers. Finally, Section 6.5 studies KPP front-speed estimation in both 2D and 3D time-dependent flows, evaluating invariant measures and eigenvalue convergence to illustrate how the MF+DP warm start improves downstream Feynman–Kac estimators beyond pure distributional metrics.

### 6.1 4D KELLER–SEGEL SYSTEM EXTENSIONS

To assess whether our two–stage pipeline scales beyond three spatial dimensions, we extend the Keller–Segel dynamics in (15) to a four–dimensional state variable $x = (x_1, x_2, x_3, x_4) \in \mathbb{R}^4$. The density $\rho(t, x)$ satisfies the same chemotactic PDE,

$$\partial_t \rho = \mu \Delta \rho + \chi \nabla_x \cdot (\rho \nabla_x (K * \rho)),$$

with identical regularization and parameters as in the 3D laminar experiment in Section 4. The background velocity is now a 3D laminar profile embedded in four dimensions:

$$v(x; \sigma) = \sigma \big( e^{-(x_2^2 + x_3^2 + x_4^2)}, 0, 0, 0 \big)^\top, \tag{19}$$

so that the advection acts purely along $x_1$ while the remaining coordinates provide cross–stream structure. The interacting–particle approximation of the PDE becomes

$$dX_j = -\frac{\chi}{J} \sum_{i \neq j} \nabla K_\delta(\|X_i - X_j\|) \, dt + v(X_j; \sigma) \, dt + \sqrt{2\mu} \, dW_j, \tag{20}$$

which is the natural 4D analogue of the 3D scheme in Section 4. We implement (20) exactly as in the 3D experiment, with the only modification that initialization is uniform on the 4D unit ball.

**Evaluation metric in 4D.** Directly applying the histogram–based discrete $W_2$ used in 3D requires a $24^4$ grid (331,776 cells), and the resulting cost matrix would contain over $10^{11}$ entries, which exceeds our 480 GB RAM budget even in single precision. Therefore, for each $\sigma$ we estimate $W_2$ using *two independent subsamples* of size $m = 2000$ drawn from the reference and generated ensembles. We compute the Earth–Mover's Distance (EMD) between these point clouds with a squared Euclidean cost. This modification affects only the evaluation metric; both MeanFlow and DP training remain unchanged and identical to the 3D setting.

**Results.** The left panel of Fig. 5 summarizes the approximate $W_2$ values for MeanFlow and our DP refinement across $\sigma \in \{20, \dots, 200\}$. In the moderate–advection range ($\sigma \leq 100$), both methods achieve errors around $4 \times 10^{-3}$, with DP consistently slightly better. As the system enters the singular–perturbation regime, MeanFlow's error grows sharply while DP remains nearly flat: at $\sigma = 120$, MeanFlow jumps to $0.0572$ whereas DP stays at $0.0036$ (about $16\times$ smaller); at $\sigma = 160$–$180$, the gap grows to $20\times$–$26\times$; even at $\sigma = 200$, MeanFlow reaches $0.1526$ while DP remains $0.0134$. The right panel of Fig. 5 illustrates the blow–up of MeanFlow beyond $\sigma \approx 120$, contrasted with the stability of DP.

These results show that *the advantage of our two–stage design becomes even more pronounced in higher dimensions*. Stage I (MeanFlow) provides a fast global displacement that moves particles near the correct 4D support; Stage II then performs a lightweight, geometry–aware OT refinement that remains tractable on small subsamples yet robustly prevents the high–dimensional misalignment suffered by the one–step generator. This behavior is consistent with the intuition established in the 3D analysis and further validates our claim that the "coarse–to–fine" MeanFlow+DP strategy is well suited for challenging high–dimensional transport problems.

| $\sigma$ | Meanflow | DP refinement |
|---|---|---|
| 20 | 0.0038 | 0.0030 |
| 40 | 0.0040 | 0.0031 |
| 60 | 0.0040 | 0.0031 |
| 80 | 0.0039 | 0.0032 |
| 100 | 0.0044 | 0.0039 |
| 120 | 0.0572 | 0.0036 |
| 140 | 0.1011 | 0.0038 |
| 160 | 0.1129 | 0.0043 |
| 180 | 0.1431 | 0.0066 |
| 200 | 0.1526 | 0.0134 |

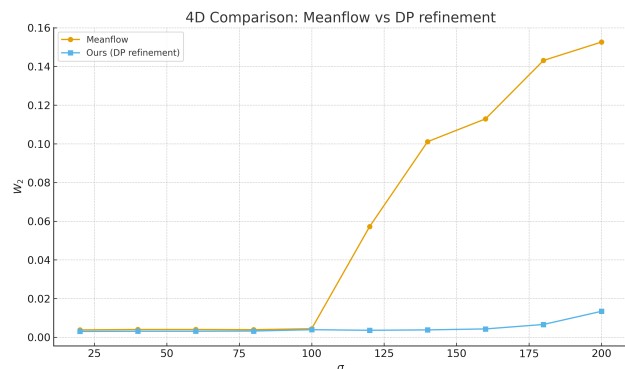

Figure 5: 4D Keller–Segel with laminar flow. Left: approximate $W_2$ values for MeanFlow and DP refinement across $\sigma$. Right: $W_2$ vs. $\sigma$ curves, showing MeanFlow's error rapidly increasing in the stiff advection regime while DP remains stable.

## 6.2 ABLATION STUDY: BATCH SIZE AND COUPLING REFRESH RATE

To assess the efficiency–accuracy trade-offs in Stage-II Wasserstein refinement, we perform an ablation study over two key hyperparameters: the DP mini-batch size $B_{\mathrm{dp}}$ and the coupling re-

fresh period $\gamma_{\text{renew}}$. Table 1 reports the resulting approximate Wasserstein-2 distances across $\sigma \in \{20, \ldots, 200\}$, along with wall-clock time and peak memory usage.

**Effect of DP mini-batch size.** With $\gamma_{\text{renew}} = 50$ fixed, varying $B_{\text{dp}}$ from 1500 to 3500 results in only minor fluctuations in final $W_2$ values, where differences are typically on the order of $10^{-3}$ and no configuration dominates uniformly. However, the computational cost scales nearly linearly with batch size, with total runtime increasing from 893 s at 1500 to more than 3800 s at 3500, while peak memory remains unchanged. Thus, a small batch size significantly accelerates training without sacrificing performance or introducing memory overhead. We therefore adopt $B_{\text{dp}} = 1500$ (bold column in Table 1) as our default configuration.

**Effect of coupling refresh rate.** Fixing the batch size to 1500, we vary $\gamma_{\text{renew}} \in \{5, 25, 50, 100\}$. Very frequent refreshes ($\gamma_{\text{renew}} = 5$) provide only negligible $W_2$ gains while being $\sim 4\times$ slower, whereas overly sparse refreshes ($\gamma_{\text{renew}} = 100$) degrade accuracy, especially at larger $\sigma$. The intermediate value $\gamma_{\text{renew}} = 50$ achieves the best balance between stability and efficiency, reducing runtime to 854 s while attaining nearly the lowest $W_2$ across the sweep, with identical memory usage to all other settings. We therefore use $\gamma_{\text{renew}} = 50$ (bold column in Table 1) for all main experiments, as it offers a substantial reduction in runtime while preserving accuracy and memory consumption.

Table 1: DP $W_2$ under different batch sizes and coupling refresh rates. Bold columns indicate the configuration adopted in the main experiments.

| | Batch size sweep ($\gamma_{\text{renew}} = 50$) | | | | | Coupling refresh sweep (batch size = 1500) | | | |
| --- | --- | --- | --- | --- | --- | --- | --- | --- | --- |
| $\sigma$ | **1500** | 2000 | 2500 | 3000 | 3500 | 5 | 25 | **50** | 100 |
| 20 | **0.0047** | 0.0039 | 0.0044 | 0.0039 | 0.0039 | 0.0038 | 0.0039 | **0.0047** | 0.0081 |
| 40 | **0.0046** | 0.0038 | 0.0039 | 0.0040 | 0.0043 | 0.0043 | 0.0045 | **0.0046** | 0.0073 |
| 60 | **0.0049** | 0.0045 | 0.0040 | 0.0043 | 0.0043 | 0.0045 | 0.0040 | **0.0049** | 0.0090 |
| 80 | **0.0056** | 0.0042 | 0.0043 | 0.0047 | 0.0042 | 0.0046 | 0.0045 | **0.0056** | 0.0105 |
| 100 | **0.0059** | 0.0054 | 0.0050 | 0.0048 | 0.0045 | 0.0049 | 0.0050 | **0.0059** | 0.0115 |
| 120 | **0.0065** | 0.0053 | 0.0051 | 0.0056 | 0.0053 | 0.0054 | 0.0055 | **0.0065** | 0.0125 |
| 140 | **0.0073** | 0.0053 | 0.0053 | 0.0058 | 0.0056 | 0.0052 | 0.0058 | **0.0073** | 0.0111 |
| 160 | **0.0082** | 0.0064 | 0.0060 | 0.0058 | 0.0059 | 0.0064 | 0.0064 | **0.0082** | 0.0133 |
| 180 | **0.0140** | 0.0100 | 0.0095 | 0.0100 | 0.0102 | 0.0102 | 0.0094 | **0.0140** | 0.0169 |
| 200 | **0.0214** | 0.0166 | 0.0171 | 0.0172 | 0.0190 | 0.0163 | 0.0160 | **0.0214** | 0.0202 |
| time (s) | **893.2** | 1395.0 | 2089.1 | 2892.9 | 3830.4 | 3300.4 | 1109.5 | **854.7** | 771.5 |
| memory (MB) | **1154.7** | 1154.7 | 1154.7 | 1154.7 | 1154.7 | 1154.7 | 1154.7 | **1154.7** | 1154.7 |

## 6.3 Comparison with other baselines

**IMM (self distillation) Zhou et al. (2025).** We implement an inductive moment matching (IMM) style one-step baseline adapted to the Keller Segel setting and closely related to the self-distilled flow-map view and IMM framework of Zhou et al. (2025). and Boffi et al. (2025). For each physical parameter $\sigma$ on the same 8-point logarithmic grid used by MeanFlow, we generate paired endpoints $(x_0, x_1)$ by simulating the KS dynamics from the shared initial configuration $x_0$ to the terminal state $x_1$. The model is a compact MLP (width 64, depth 4) that shares both architecture and $\sigma$-conditioning with the MeanFlow network and parameterizes a time-conditioned flow map

$$f_\theta(x, t \to s, \log_{10} \sigma) = x + (s - t)\, g_\theta(x, t, s, \log_{10} \sigma), \tag{21}$$

so that a single forward pass predicts the displacement from any source time $t$ to any earlier target time $s$. During training we sample a mini-batch of KS pairs for a randomly chosen $\sigma$ and draw nested times $0 \le s < r < t \le 1$. We define a linear path between data and prior, $x_t = (1-t)x_1 + tx_0$ and $x_r = (1-r)x_1 + rx_0$, and enforce temporal consistency by matching the distributions of the "big-step" map $f_\theta(x_t, t \to s)$ and the "small-step" map $f_\theta(x_r, r \to s)$. These two sets of samples are partitioned into groups and compared using a Laplacian-kernel MMD loss whose bandwidth scales with the step size $|t - s|$, yielding the IMM-style objective

$$\mathcal{L}_{\text{IMM}}(\theta) = \mathbb{E}_{\sigma, (x_0, x_1), s < r < t}\left[\text{MMD}^2\big(f_\theta(x_t, t \to s), f_\theta(x_r, r \to s)\big)\right]. \tag{22}$$

Gradients flow only through the big-step branch while the midpoint branch is treated as a frozen teacher, implementing a self-distillation scheme without an external diffusion teacher. All hyperparameters are kept identical to the MeanFlow stage to ensure a fair comparison.

**SF2M-OT Tong et al. (2023b).** As an OT-style continuous-time benchmark, we adapt a state-free flow matching (SF2M) method to the Keller Segel (KS) system. For each $\sigma$ in the same training grid as our DP experiments, we build a dataset of endpoint pairs by initializing particles on the unit sphere and evolving them with the Keller Segel simulator, yielding triples $(x_0, x_1, \log_{10} \sigma)$. The SF2M model predicts a velocity field $v_\theta(x, t, \log_{10} \sigma)$ and a score field $s_\theta(x, t, \log_{10} \sigma)$ from the concatenated input $[x, \log_{10} \sigma, t]$. Training follows a Brownian-bridge construction: given a random time $t \in (0, 1)$, we form the conditional bridge mean $m_t = (1 - t)x_0 + tx_1$ and variance $k_t = t(1 - t)$, sample noisy bridge states $x_t \sim \mathcal{N}(m_t, \sigma_b^2 k_t I)$, and compute the analytic bridge score $\nabla_x \log p_t(x_t \mid x_0, x_1)$ and probability-flow velocity $u_t(x_t \mid x_0, x_1)$. The loss is a sum of squared errors between network predictions and these closed-form targets,

$$\mathcal{L}_{\mathrm{SF2M}}(\theta) = \mathbb{E}\Big[\|v_\theta(x_t, t, \ell) - u_t\|^2 + \lambda_{\mathrm{score}} \|s_\theta(x_t, t, \ell) - \nabla_x \log p_t\|^2\Big], \tag{23}$$

with bridge noise scale $\sigma_b = 0.4$ and $\lambda_{\mathrm{score}} = 1$. At test time, SF2M defines a simulation-free ODE initialized at prior samples $x_0$; integrating $\dot{x}_t = v_\theta(x_t, t, \ell)$ from $t = 0$ to $1$ with 100 Euler steps yields samples whose endpoints approximate the KS distribution for the given $\sigma$. Because the training path is a Brownian bridge between $(x_0, x_1)$, this baseline can be interpreted as an OT-informed Schrödinger-bridge approximation that directly targets transport between prior and terminal KS states.

**Experiment analysis.** Table 2 and Figure 8 compare MeanFlow, IMM (self distillation), SF2M-OT, and our DP refinement across $\sigma \in \{20, \ldots, 200\}$ using the approximate 2-Wasserstein metric $W^2$. In the moderate-advection regime ($\sigma \leq 100$), all one-step models achieve similar accuracy, with the OT-based baselines (IMM and SF2M-OT) generally outperforming MeanFlow and DP refinement providing the best overall performance. As $\sigma$ increases into the singular-perturbation regime, the error of MeanFlow grows rapidly, whereas IMM and SF2M-OT remain substantially more stable; DP refinement consistently attains the lowest $W^2$ across the entire range. Figures 6 and 7 further illustrate these trends via qualitative histograms in an interpolation case ($\sigma = 20$) and an extrapolation case ($\sigma = 160$), where our method produces samples that most closely match the KS reference distributions.

Table 2: Approximate 2-Wasserstein metric $W_2$ for different baselines on the 3D KS–Laminar experiment.

| $\sigma$ | Meanflow | IMM (self distillation) | SF2M-OT | DP refinement |
|---|---|---|---|---|
| 20 | 0.0084 | 0.0089 | 0.0051 | **0.0047** |
| 40 | 0.0068 | 0.0096 | 0.0046 | **0.0046** |
| 60 | 0.0070 | 0.0094 | 0.0056 | **0.0049** |
| 80 | 0.0105 | 0.0103 | 0.0059 | **0.0056** |
| 100 | 0.0132 | 0.0115 | 0.0065 | **0.0059** |
| 120 | 0.0201 | 0.0117 | 0.0074 | **0.0065** |
| 140 | 0.0216 | 0.0120 | 0.0091 | **0.0073** |
| 160 | 0.0403 | 0.0145 | 0.0097 | **0.0082** |
| 180 | 0.0832 | 0.0363 | 0.0186 | **0.0140** |
| 200 | 0.1970 | 0.0897 | 0.0403 | **0.0214** |

Figure 6: Qualitative comparison of 2D histograms at $\sigma = 20$ (interpolation regime). From top to bottom we show MeanFlow, DP refinement, IMM (self distillation), SF2M-OT, and the KS reference solution. All methods produce roughly isotropic particle clouds, but MeanFlow exhibits slightly blurred density and a noticeable mismatch in the outer mass. IMM and SF2M-OT reduce these distortions, while DP refinement yields the sharpest and most symmetric match to the reference across all coordinate projections.

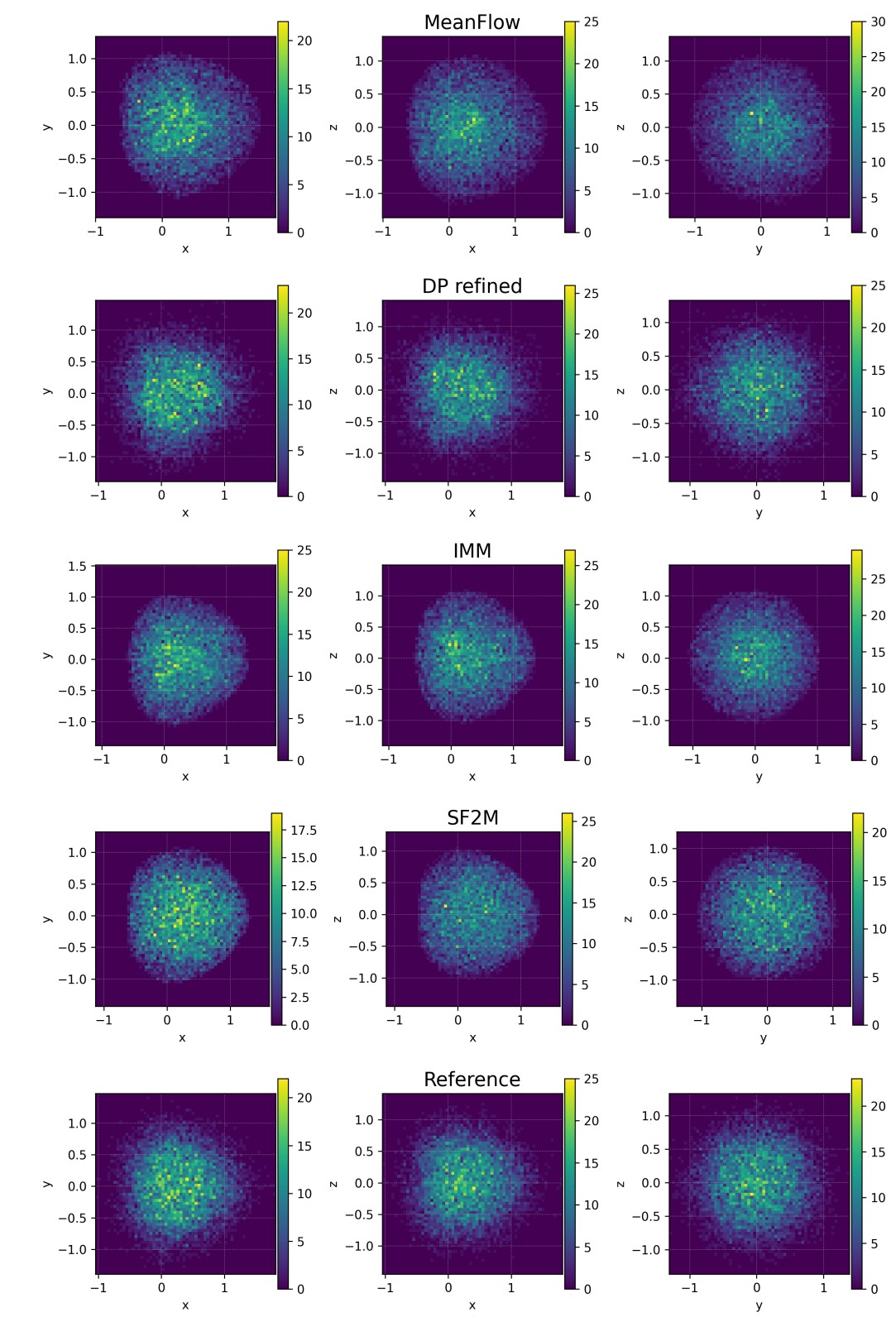

Figure 7: Qualitative comparison of 2D histograms at $\sigma = 160$ (extrapolation regime). MeanFlow significantly distorts the particle distribution, with mass shifted away from the reference and visible anisotropy. IMM and SF2M-OT partially correct these errors but still exhibit broader, less concentrated densities. In contrast, DP refinement closely tracks the reference histograms in all views, preserving both the core concentration and the shape of the support, confirming that it generalizes best beyond the training range.

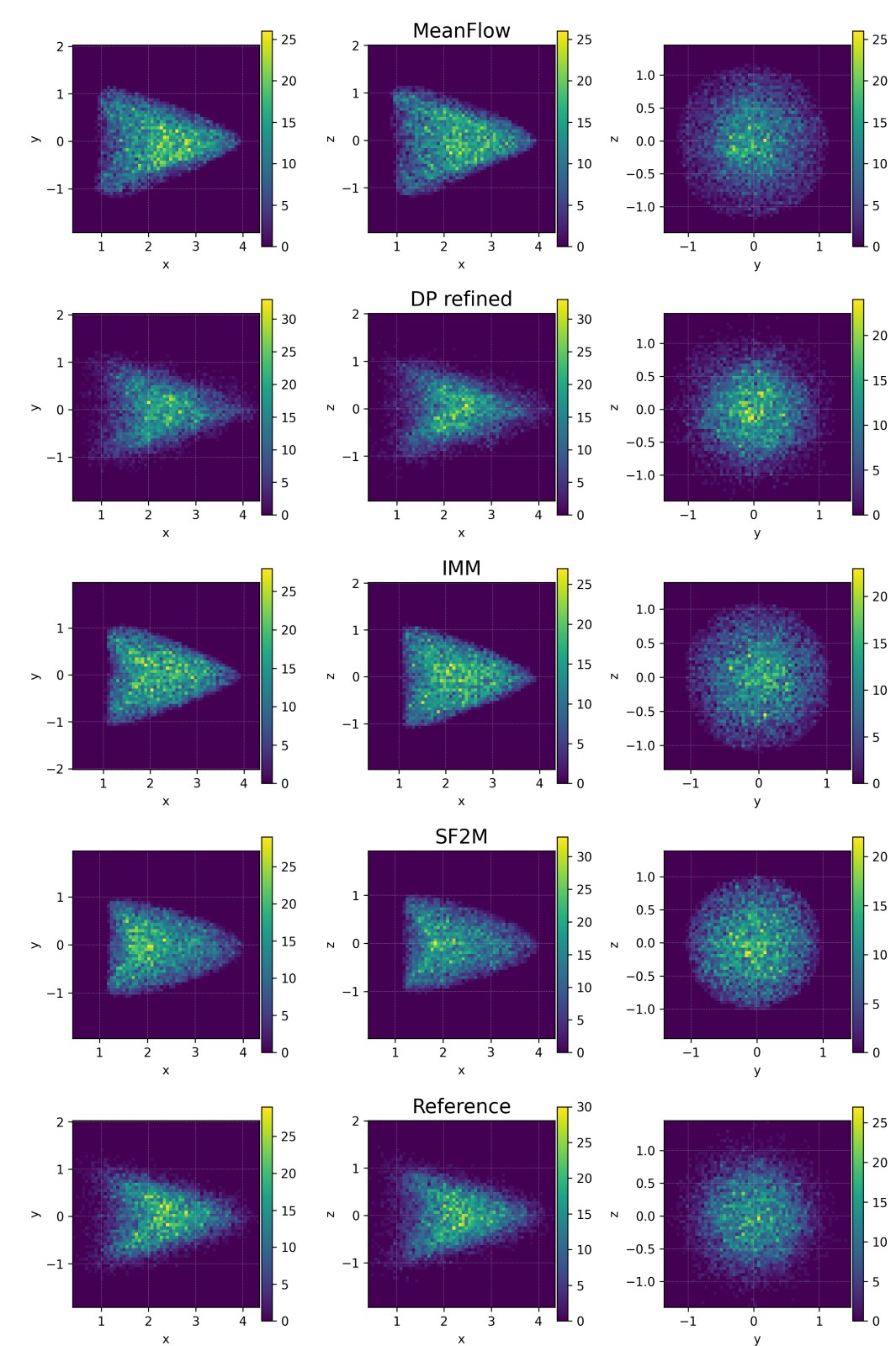

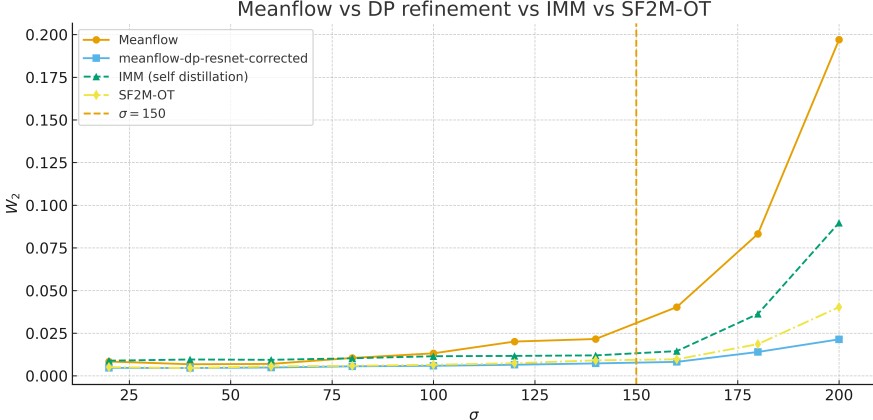

Figure 8: Comparison of the 2-Wasserstein metric $W_2$ across $\sigma$ for MeanFlow, DP refinement, IMM (self distillation), and SF2M-OT.

## 6.4 COMPARISON WITH REGULARIZED SINKHORN

In this section, we compare our exact mini-batch EMD refinement against entropically regularized Sinkhorn solvers using coefficients $\varepsilon \in \{0.05, 0.01, 0.005, 0.001\}$. We keep all components of the two-stage pipeline fixed and only change the OT solver in Stage II. Table 3 reports the resulting approximate $W_2$ across $\sigma \in \{20, \dots, 200\}$ along with runtime and memory usage, while Figure 9 provides qualitative comparisons of the learned particle distributions under each regularization strength.

Across all coefficients, Sinkhorn achieves reasonable alignment but exhibits a clear accuracy–efficiency trade-off. Larger entropic regularization ($\varepsilon = 0.05$ and $\varepsilon = 0.01$) leads to excessive smoothing of the transport plan, producing visibly blurred distributions and noticeably higher $W_2$ values, especially in the singular-perturbation regime ($\sigma \geq 150$). Reducing the coefficient improves accuracy, and $\varepsilon = 0.001$ performs closest to EMD. However, the computational cost increases dramatically as $\varepsilon$ becomes small: the total runtime grows from 6538 s at $\varepsilon = 0.05$ to more than $6.2 \times 10^4$ s at $\varepsilon = 0.001$, while memory usage remains higher than that of EMD.

In contrast, EMD refinement simultaneously achieves: (i) the lowest or near-lowest $W_2$ across all $\sigma$; (ii) an order-of-magnitude speedup compared to even the fastest Sinkhorn run; (iii) the smallest memory footprint. These results indicate that, after MeanFlow has concentrated mass on the correct support, the OT geometry becomes sufficiently well-conditioned that exact EMD is both more accurate and dramatically more efficient than Sinkhorn. We therefore adopt EMD as the default mini-batch OT solver in all main experiments.

Table 3: Comparison of $W_2$ under different regularized Sinkhorn coefficients. EMD refinement is both significantly faster and yields the best overall accuracy.

| $\sigma$ | 0.001 | 0.005 | 0.01 | 0.05 | EMD refinement |
|---|---|---|---|---|---|
| 20 | 0.003961 | 0.004436 | 0.004859 | 0.007994 | 0.0047 |
| 40 | 0.004126 | 0.004357 | 0.004747 | 0.007871 | 0.0046 |
| 60 | 0.003915 | 0.004419 | 0.004765 | 0.008195 | 0.0049 |
| 80 | 0.004640 | 0.005143 | 0.005538 | 0.009247 | 0.0056 |
| 100 | 0.004764 | 0.004913 | 0.006174 | 0.009604 | 0.0059 |
| 120 | 0.005479 | 0.005843 | 0.007758 | 0.010074 | 0.0065 |
| 140 | 0.005214 | 0.006203 | 0.008551 | 0.010624 | 0.0073 |
| 160 | 0.006398 | 0.006902 | 0.009029 | 0.011593 | 0.0082 |
| 180 | 0.010744 | 0.012127 | 0.020977 | 0.025443 | 0.0140 |
| 200 | 0.015145 | 0.017245 | 0.027149 | 0.039835 | 0.0214 |
| time (s) | 62879.88 | 44889.60 | 26305.99 | 6537.78 | 893.2 |
| memory (MB) | 1479.76 | 1445.29 | 1504.32 | 1442.92 | 1154.7 |

Figure 9: Qualitative comparison of Mini-batch OT refinement under different Sinkhorn regularization coefficients versus exact EMD refinement when $\sigma = 160$. Large coefficients oversmooth the distribution; small coefficients improve accuracy but become extremely slow. EMD achieves the cleanest alignment at the lowest computational cost.

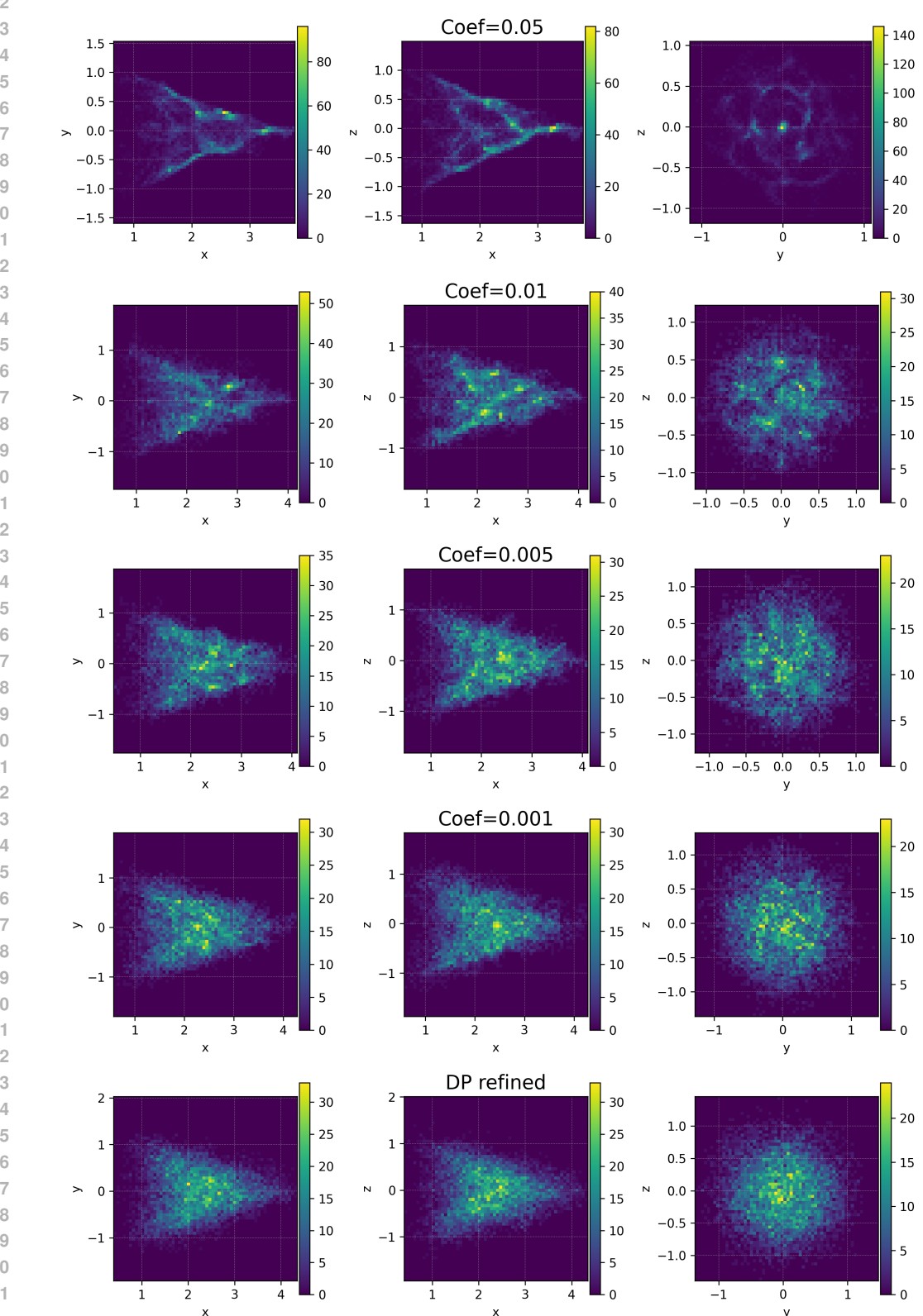

## 6.5 KOLMOGOROV-PETROVSKY-PISKUNOV (KPP) FRONT SPEED EXPERIMENT

### 6.5.1 KPP FRONT SPEED EXPERIMENT

Front propagation in fluid flows becomes a hot topic in the physical sciences Xin (2009). The reaction-diffusion-advection equation with Kolmogorov-Petrovsky-Piskunov (KPP) nonlinearity is as follows Kolmogorov et al. (1937):

$$u_t + \mathbf{v}(x)\cdot\nabla u = \kappa\Delta u + r\,u(1-u), \tag{24}$$

where $\kappa$ is diffusion constant, $\mathbf{v}$ is an incompressible velocity field , and $u$ is the concentration of reactant. If the velocity field is $T$-periodic in space and time, the minimal front speed satisfies the variational formula Nolen et al. (2005):

$$c^*(e) = \inf_{\lambda>0} \frac{\mu(\lambda)}{\lambda} \tag{25}$$

where $\mu(\lambda)$ is the principal eigenvalue of parabolic operator $\partial_t - \mathcal{A}$ with:

$$\mathcal{A}w := \kappa\Delta_x w + \left(2\kappa\lambda\,\mathbf{e}_1 + \mathbf{v}\right)\cdot\nabla_x w + \left(\kappa\lambda^2 + \lambda\,\mathbf{v}\cdot\mathbf{e}_1 + 1\right)w \tag{26}$$

For a fixed $\lambda > 0$, the twisted diffusion on $\mathbb{T}^2 = [0, 2\pi)^2$ is

$$dX_t = \left(2\kappa\lambda\,\mathbf{e}_1 + \mathbf{v}(X_t, t)\right)dt + \sqrt{2\kappa}\,dW_t \tag{27}$$

with

$$\mathbf{v}(x,t) = \begin{pmatrix} -\cos(x_2) - \theta\cos(2\pi t)\,\sin(x_1) \\ \cos(x_1) + \theta\cos(2\pi t)\,\sin(x_2) \end{pmatrix}, \tag{28}$$

which corresponds to the operator in 26. The Feynman–Kac semigroup uses the potential

$$V_\lambda(x,t) = \kappa\lambda^2 + \lambda\,\mathbf{v}(x,t)\cdot\mathbf{e}_1, \tag{29}$$

So that, with one forcing period normalized to $\Delta T = 1$ and each period discretized into $r_n = 2^{-\log_2 \Delta t}$ substeps of size $\Delta t = 1/r_n$, we can firstly do the Euler–Maruyama propagation of (27) with multiplicative weights

$$w_j^i = \exp\left(V_\lambda(X_j^i, t_j)\Delta t\right) \tag{30}$$

with $\overline{w}_j = \frac{1}{M}\sum_i w_j^i$. Finally, a multinomial resampling $\propto \{w_j^i\}$ approach is used in each substep. The running estimators are

$$\widehat{\mu}_T(\lambda) = \kappa + \frac{1}{T}\sum_{t=1}^{Tr_n}\log\overline{w}_t, \qquad \widehat{c}_T(\lambda) = \widehat{\mu}_T(\lambda)/\lambda, \tag{31}$$

Meanwhile, let $\gamma_n(\varphi) = \mathbb{E}[\varphi(X_n)\prod_{t=1}^n \overline{w}_t]$ and $\eta_n = \gamma_n/\gamma_n(1)$ be the normalized empirical measure after resampling. Then $\eta_n \Rightarrow \eta_\lambda^\star$ and $\frac{1}{n}\log\gamma_n(1) \to \mu(\lambda)$ as $n \to \infty$. Consequently, we approximate $\eta_\lambda^\star$ by running the FK particle system long enough and collecting the terminal cloud. In this setting, we start from the uniform distribution on $[0, 2\pi)^2$ and uses a large $T = 4096$.

**Experimental Setup**   We study KPP front–speed estimation on the two–torus $\mathbb{T}^2 = [0, 2\pi)^2$ under a time–periodic cellular flow. We fix $(\lambda, \theta) = (2.0, 1.0)$ and discretize each forcing period to $r_n = 256$ substeps ($\log_2 \Delta t = -8$). Each run uses $M = 2 \times 10^4$ particles; we print $\widehat{c}_T(\lambda) = \widehat{\mu}_T(\lambda)/\lambda$ at dyadic generations $T = 1, 2, 4, \ldots$. Two initializations are compared: *warm* (Meanflow with DP refinement; Meanflow samples from a Gaussian source with std $\approx \pi$ and a 6-block residual corrector on the torus refines it, both conditioned on $\sigma$) and *cold* (Uniform on $[0, 2\pi)^2$). The $sigma$ is defined as the diffusion constant $\kappa$ in 26 in this scenario. We use dyadic diffusion constant levels $\sigma = 2^\ell$ with uniformly spaced exponents for training

$$\ell_k^{\mathrm{tr}} = -2 - (k-1)\cdot 0.25, \qquad k = 1, \ldots, 8, \tag{32}$$

and testing:

$$\ell_k^{\mathrm{te}} = -2 - (k-1)\cdot 0.25, \qquad k = 1, \ldots, 11. \tag{33}$$

with the two metrics: (i) *Eigenvalue convergence*: curves of $\widehat{c}_T(\lambda)$ vs. $T$ for warm (meanflow with DP refinement) vs. cold start. (ii) *Invariant-measure accuracy*: histogram Wasserstein-2 on the torus (periodic ground metric, $72\times72$ bins) against a long-horizon FK reference.

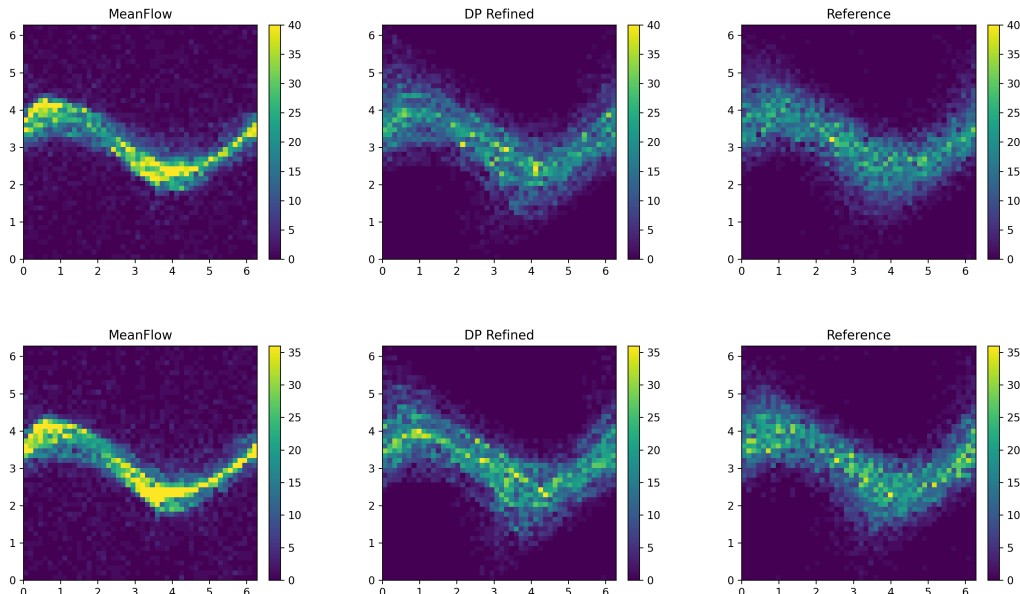

Figure 10: Empirical invariant measures on $\mathbb{T}^2$ across diffusion constants. Rows correspond to $\sigma = 2^{-3.75}$ (top, *within the training range*) and $\sigma = 2^{-4}$ (bottom, *extrapolation beyond training*); columns show MeanFlow (left), MeanFlow+DP refinement (middle), and the resolved FK reference (right). Each panel is a $72 \times 72$ histogram on $[0, 2\pi)^2$. The DP corrector contracts spurious mass and sharpens the anisotropic ridge, bringing the warm-start distribution visibly closer to the reference at both $\sigma$, including the extrapolation case.

| $-\log_2 \sigma$ | **Meanflow** | **DP refinement** |
|---|---|---|
| 2.00 | 0.04766 | **0.01376** |
| 2.25 | 0.04755 | **0.01393** |
| 2.50 | 0.05325 | **0.01239** |
| 2.75 | 0.05918 | **0.01228** |
| 3.00 | 0.06385 | **0.01057** |
| 3.25 | 0.07173 | **0.01477** |
| 3.50 | 0.08704 | **0.02003** |
| 3.75 | 0.07726 | **0.01540** |
| 4.00 | 0.10120 | **0.02159** |
| 4.25 | 0.11950 | **0.01539** |
| 4.50 | 0.11760 | **0.04559** |

Figure 11: *Left:* $W_2$ (lower is better) between the empirical measure and the long-$T$ reference for MF and the DP refinement, evaluated at $\sigma = 2^{-\ell}$. *Right:* The same numbers plotted versus $-\log_2 \sigma$; the green dashed line marks the end of the *training* range ($-\log_2 \sigma = 3.75$), so points to the right are *extrapolation*. Across the entire grid, including the extrapolation regime, the DP refinement uniformly reduces error relative to MF and flattens the growth of $W_2$.

**Experimental Analysis** **Qualitative agreement of invariant measures.** Figure 10 compares empirical invariant measures on $\mathbb{T}^2$ for two representative diffusion constant: $\sigma = 2^{-3.75}$ (within training) and $\sigma = 2^{-4.25}$ (extrapolation). Across both rows the MeanFlow+DP (middle) corrects the one–shot MeanFlow bias (left) by contracting spurious mass near the central trough and sharpening the anisotropic ridge that aligns with the cellular advection; the resulting density closely matches the long–run FK reference (right). Notably, the same geometric improvement persists at $\sigma = 2^{-4.25}$, evidencing strong out–of–range generalization.

**Quantitative sampling accuracy across $\sigma$.** Figure 11 summarizes the 2-Wasserstein distance on the torus between the empirical measure and the long-$T$ reference. The DP refinement uni-

| iteration | MF-only init | MF+DP init |
|---|---|---|
| 1 | 0.733899 | 0.761573 |
| 2 | 0.763092 | 0.773892 |
| 4 | 0.770337 | 0.788397 |
| 8 | 0.795811 | 0.781912 |
| 16 | 0.784826 | 0.785587 |
| 32 | 0.787654 | 0.783146 |
| 64 | 0.787575 | 0.775075 |
| 128 | 0.783591 | 0.779001 |

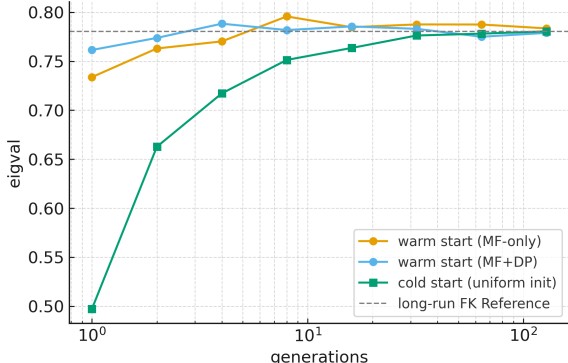

Figure 12: Eigenvalue convergence with warm and cold starts. The plot shows $\hat{c}_T(\lambda) = \hat{\mu}_T(\lambda)/\lambda$ versus generations $T$ (log scale) for warm (MF-only), warm (MF+DP), and cold (uniform) initializations. The dashed line is the long-run Feynman–Kac (FK) reference. The left table reports the estimator at early iterations $T \in \{1, 2, 4, \ldots, 128\}$ for the two warm starts.

formly lowers $W_2$ over the entire grid, including extrapolation points (right panel). For example, at $-\log_2 \sigma = 3.0$ the error drops from $0.06385$ (MF) to **0.01057** (MF+DP), a relative reduction of $\sim 83\%$; at $-\log_2 \sigma = 4.0$ it drops from $0.10120$ to **0.02159** ($\sim 79\%$ reduction); and at the extrapolation point $-\log_2 \sigma = 4.25$ it drops from $0.11950$ to **0.01539** ($\sim 87\%$ reduction). The left table in Fig. 11 shows consistent improvements at all reported $\sigma$s; the right plot visualizes the same trend and marks the boundary of the training range ($-\log_2 \sigma = 3.75$) with a dashed line.

Initial estimate nearly matches the long-run reference. Figure 12 reports $\hat{c}_T(\lambda) = \hat{\mu}_T(\lambda)/\lambda$ versus generations $T$ for the warm (MF+DP) and cold (uniform) starts. With DP refinement, the very first estimate ($T = 1$) is already almost identical to the long-run FK reference and remains close thereafter, indicating minimal transient bias and variance. The MF-only warm start begins slightly below the reference but converges within a few generations. In contrast, the cold start begins far from the limit and requires roughly an order of magnitude more generations to catch up. Overall, DP refinement provides the strongest warm start: its initial eigenvalue essentially equals the long-run FK reference, and it stabilizes fastest.

### 6.5.2 3D KPP Front Speed Experiment with time-dependent Kolmogorov flow

In this section, we study our method on the 3D KPP equation:

$$u_t + \mathbf{v}(\mathbf{x}) \cdot \nabla u = \kappa \Delta u + r\, u(1 - u), \tag{34}$$

with a three-dimensional time-dependent Kolmogorov flow velocity field:

$$\mathbf{v}(\mathbf{x}, t) = \big(\sin(x_3 + \sin(2\pi t)), \sin(x_1 + \sin(2\pi t)), \sin(x_2 + \sin(2\pi t))\big) \tag{35}$$

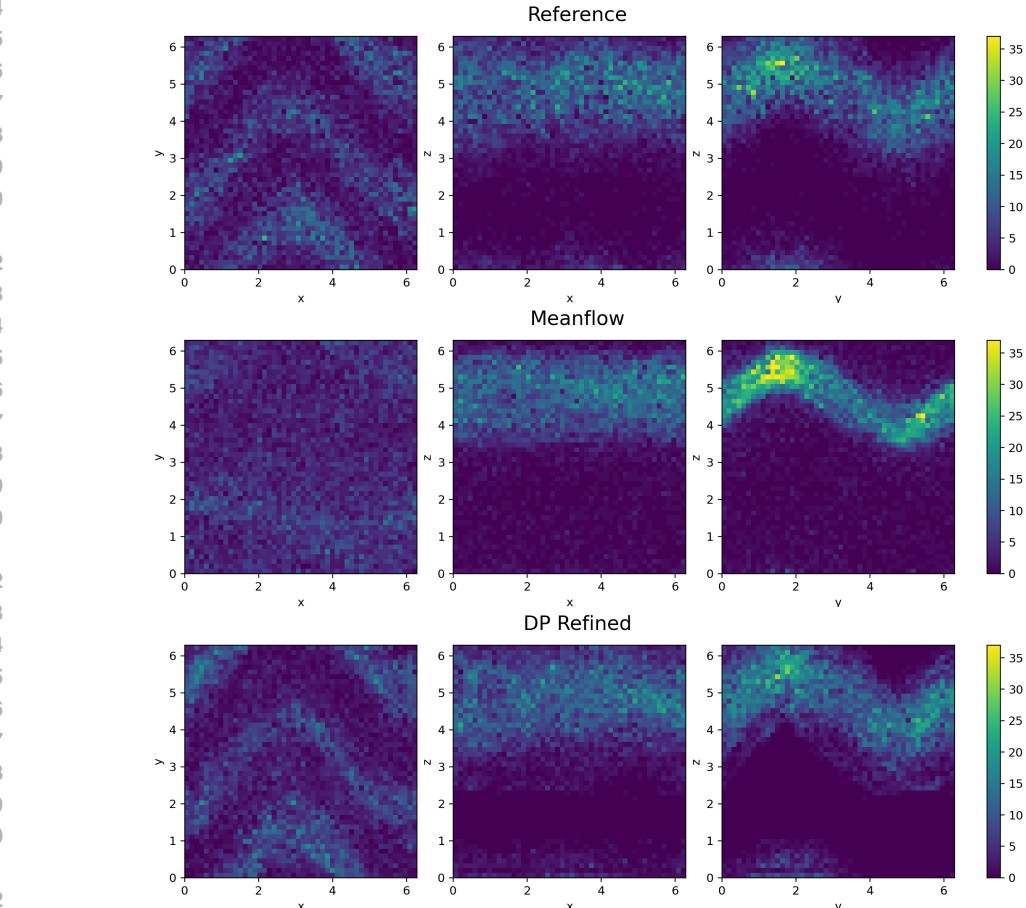

Figure 13: Qualitative comparisons of $(x, y)$, $(x, z)$, $(y, z)$ planes at $\sigma = 2^{-4.5}$ for the 3D KPP system in 3D time-dependent Kolmogorov flow. (a) Reference solution projected to three coordinate planes. (b) Predicted solution projected to the three coordinate planes by MeanFlow (c) DP Refinement solution projects to the three coordinate planes.

| $-\log_2 \sigma$ | Meanflow | DP refinement |
|---|---|---|
| 2.00 | 0.15910 | **0.01598** |
| 2.25 | 0.12570 | **0.02298** |
| 2.50 | 0.14420 | **0.01927** |
| 2.75 | 0.16460 | **0.02803** |
| 3.00 | 0.12500 | **0.03655** |
| 3.25 | 0.11750 | **0.03141** |
| 3.50 | 0.13420 | **0.03403** |
| 3.75 | 0.12610 | **0.02526** |
| 4.00 | 0.10490 | **0.06062** |
| 4.25 | 0.13250 | **0.02815** |
| 4.50 | 0.10990 | **0.02098** |

Figure 14: *Left:* $W_2$ (Projection on the xz plane) between the empirical measure and the long-$T$ reference for MF and the DP refinement, evaluated at $\sigma = 2^{-\ell}$. *Right:* The same numbers plotted versus $-\log_2 \sigma$; the green dashed line marks the end of the *training* range ($-\log_2 \sigma = 3.75$), so points to the right are *extrapolation*. Across the entire grid, including the extrapolation regime, the DP refinement uniformly reduces error relative to MF and flattens the growth of $W_2$.

| iteration | MF-only init | MF+DP init |
|---|---|---|
| 1 | 0.566547 | 0.613531 |
| 2 | 0.551988 | 0.610457 |
| 4 | 0.546482 | 0.621819 |
| 8 | 0.576172 | 0.599699 |
| 16 | 0.597887 | 0.590171 |
| 32 | 0.609560 | 0.614999 |
| 64 | 0.609690 | 0.615682 |
| 128 | 0.614481 | 0.613865 |

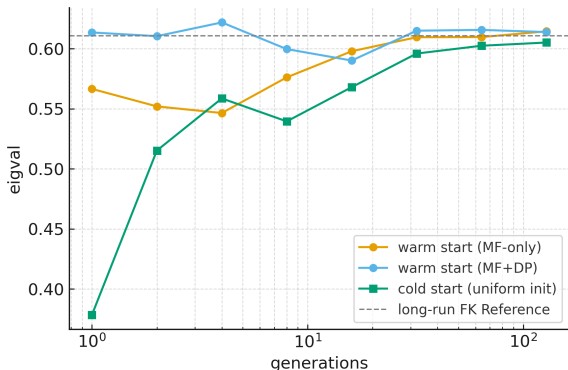

Figure 15: **Eigenvalue convergence with warm and cold starts.** The plot shows $\hat{c}_T(\lambda) = \hat{\mu}_T(\lambda)/\lambda$ versus generations $T$ (log scale) for **warm (MF-only)**, **warm (MF+DP)**, and **cold (uniform)** initializations. The dashed line is the long-run Feynman–Kac (FK) reference. The left table reports the estimator at early iterations $T \in \{1, 2, 4, \ldots, 128\}$ for the two warm starts.

Experimental Setup The setup parallels that of the 2D experiment. The spatial domain is $\mathbb{T}^3 = [0, 2\pi)^3$, with the time period normalized to $T = 1$. We employ Euler–Maruyama integration with $\Delta t = 2^{-8}$ and perform FK propagation using $2 \times 10^4$ particles per diffusion level.

The diffusion constant $\sigma$ (identical to $\kappa$) follows a dyadic grid $\sigma = 2^{-\ell}$. Training levels are defined for $-\log_2 \sigma \leq 3.75$, and evaluation extends up to $-\log_2 \sigma = 4.5$ to test out-of-range generalization. Furthermore, we compare three initialization modes, where the first is the *Cold start*, where we start from the uniform distribution on $\mathbb{T}^3$ and this case served as the reference group. Meanwhile, we tested the two warm start methods through meanflow and our two-step diffusion pipeline. where we would like to test our model on the following two metrics:

1. *Eigenvalue convergence* of $\hat{c}_T(\lambda)$ versus the number of FK generations $T \in \{1, 2, 4, \ldots\}$;

2. *Invariant-measure accuracy*, where empirical particle clouds are projected onto the $(x_2, x_3)$ plane and compared with the FK reference via the 2-Wasserstein distance $W_2$.

Additionally, 2D projection visualizations (XY, XZ, YZ projections) are plotted for qualitative inspection of anisotropic structure recovery.

### 6.5.3 EXPERIMENT ANALYSIS

For the invariant measures, Figure 13 shows triptych projections at $\sigma = 2^{-4.5}$ for the FK reference (top row), MeanFlow (middle), and DP-refined outputs (bottom). Across the XY/XZ/YZ views, MeanFlow exhibits spurious mass and blurred anisotropic ridges, whereas DP refinement removes these artifacts and sharpens the cellular alignment. In the XY projection, DP restores the bifurcated high-density ridge; in XZ, it corrects the layer position and suppresses over-diffusion; and in YZ, it aligns the curved ridge with the FK reference.

Meanwhile, our pipeline achieves strong quantitative accuracy across diffusion levels. Figure 14 summarizes the $W_2$ errors over $-\log_2 \sigma \in [2.0, 4.5]$. Within the training range ($-\log_2 \sigma \leq 3.75$), DP reduces MeanFlow errors by 70–90%: for example, at $-\log_2 \sigma = 3.0$ the error decreases from 0.12500 (MF) to 0.03655 (DP), and at $-\log_2 \sigma = 2.5$ from 0.16460 to 0.02803. Beyond the training range, the method maintains strong extrapolation: at $-\log_2 \sigma = 4.25$, $W_2$ drops from 0.13250 to 0.02815, and at $-\log_2 \sigma = 4.50$ from 0.10990 to 0.02098.

Last but not least, our model exhibits rapid eigenvalue convergence. Figure 15 (and its accompanying table) reports $\hat{c}_T(\lambda)$ versus the number of FK generations. The *Warm (MF+DP)* initialization is nearly unbiased already at $T = 1$ and then remains essentially on the FK reference. The *Warm (MF-only)* initialization begins below the reference but converges within roughly 32 generations, demonstrating the stronger and more stable warm start provided by our two-step diffusion pipeline compared with MeanFlow alone.

