# OpenReview forum: "Two step diffusion:  fast sampling and reliable prediction of 3D Keller-Segel chemotaxis systems in fluid flows"
_ICLR.cc/2026/Conference — Submitted to ICLR 2026_

### Official Review · Reviewer_g1Pi · 2025-10-30

**Soundness:** 3
**Presentation:** 2
**Contribution:** 2
**Rating:** 2
**Confidence:** 2

**Summary:**

The paper proposes a 2-stage method to solve the Keller-Segel transport problem. In the first stage, solutions are generated by a MeanFlow generative model. In the second stage a corrector network is trained by optimizing a mini-batch Wasserstein objective between the generated samples and a reference distribution. The resulting method improves the generalization error across different physical parameters as demonstrated in two experiments in 2D and 3D.

**Strengths:**

- The method is sound and the corrector in the second stage achieves improved performance consistently.
- Combination of 1 NFE generative models with a final corrector step seems promising.

**Weaknesses:**

- 3D Keller-Segel chemotaxis is a very specific application. I think this framework could generalize to different problems (e.g. simulation-based inference, cell dynamics, etc.). A wider focus on the experiment/application side would make this paper more approachable for the community. It is also difficult for me to assess the experiment setup and scientific importance, since I am not familiar with Keller-Segel Chemotaxis.

- The paper contains only two main experiments. The improvements from the corrector in the second stage are evaluated and the generalization to different physical parameters $\sigma$. I would expect additional baselines to show what are the advantages of the method. Since the main goal of the paper is to map the initial particles at time 0 to particles at time 1, would it not be possible to use any framework for optimal transport as benchmarks, for example [1, 2, 3]?

- The paper relies on a 1-step generative model in stage 1. If I understand correctly, paired data $(x_1, x_2, \sigma)$ is used for training, which are generated via an interacting particle approximation. While 1-step generation is attractive, I am wondering what are the actual benefits compared to a multi-step generation with higher quality; It would be interesting to compare timings from 1 to N steps in stage 1 and consider the compute/time and quality tradeoff. In general, I think comparing the different timings: generation of data, training of models in stage 1/2, inference time is necessary to better motivate the methodology.

- Sometimes the notation is confusing, e.g. z_0 = x_1 in line 182. \citep and \citet are not used consistently. Algorithm 1, Fig. 3 and Fig. 4 take up a lot of space that could be filled with additional experiments; line 246 is confusing, "MeanFlow is restricted to mappings from a Gaussian base to the target", I thought $x_0$ was sampled from $\pi_1(\sigma)$, line 299, which I assumed was not Gaussian. Overall, I think the presentation could be improved and polished.

[1] https://proceedings.mlr.press/v238/tong24a.html

[2] https://arxiv.org/abs/2310.10649

[3] https://arxiv.org/abs/2309.16948

**Questions:**

See questions mentioned in weaknesses:

- Can you use different methods for OT and compare your method with them?
- What is the compute/time vs. quality trafeoff when you consider generative models with multiple steps in stage 1?
- Can you give timings for dataset generation/running the PDE solver compared to inference with your method?

---

> ### Author Response · Authors · 2025-11-22
>
> Thank you for your careful review and for the constructive suggestions regarding the scope of the experiments, the comparison to multi-step generative models, and the clarity of the presentation.
> We have already corrected the confusing notational choices you pointed out (e.g., the expression $z_0=x_1$ and the endpoint relations around Eqs. (12)–(14)), and we have gone through the paper to clean up inconsistent citation commands and related typos. We also clarified the description of how $x_0$ is sampled and its relation to the Gaussian base distribution.
> To address your experimental concerns, we are preparing an updated version of the manuscript to be completed by November 27, which will include:
>
> 1. OT solver ablation and OT benchmarks:
> 	Comparisons between EMD and entropy regularized Sinkhorn, ablations over coupling refresh and particle numbers;
> 	In addition, we will try different OT based benchmark methods (e.g., the optimal transport frameworks in [1, 2, 3]) and compare our two stage method against these baselines if the time is allowed.
>
> 2. Stage II scaling curves and timing breakdown:
> 	Detailed plots of runtime, memory usage, and number of OT iterations vs. batch size N;
> 3. Multi step generative models and trade off analysis:
> 	Additional experiments with multi step generative models in Stage 1 (e.g., N step MeanFlow) to study the compute/time vs. quality trade off relative to our one step + corrector design.
> 4. High dimensional generalization:
> 	Additional experiments on higher dimensional synthetic problems to illustrate how the method extends beyond the 3D Keller–Segel setting. (e.g. KPP Experiment)
> 5. Additional σ results and other experiments:
> 	Simulation results for more values of σ, will be reported in the appendix;
> 	Other one-step baselines will be reported to better highlight the advantages of our final corrector stage;
>
> We have already added an extra KPP type experiment to broaden the range of applications (this is already implemented and will be included in the revised manuscript). We hope these additions will make the advantages of the two stage approach, its trade offs in compute and accuracy, and its relation to existing OT benchmarks much clearer. Thank you again for your very helpful feedback.

---

> ### Author Response · Authors · 2025-12-03
>
> We appreciate the reviewer’s thoughtful feedback regarding scope, baselines, and compute trade-offs.
>
> (1) Broader applicability beyond the 3D KS setting.
> The appendix now includes:
>
> 4D KS with laminar flow: MeanFlow deteriorates rapidly for $\sigma > 120$, but MF+DP remains stable and low-error across all $\sigma$.
>
> KPP (2D/3D): MF+DP significantly improves invariant-measure accuracy and eigenvalue convergence.
>
> These experiments show the method’s strong generalization across different PDEs, dimensions, and flow structures.
>
> (2) Comparisons to other OT and generative baselines.
> Appendix 6.3 now includes IMM and SF2M-OT. Both baselines outperform raw MeanFlow, but MF+DP achieves the lowest $W_2$ across all regimes, particularly in stiff and extrapolation settings. Appendix 6.4 shows that EMD is dramatically faster and more accurate than Sinkhorn once MeanFlow has simplified the geometry.
>
> (3) One-step vs. multi-step generation.
> SF2M-OT uses a 100-step ODE solver at inference, but still does not match MF+DP in the stiff regime. Our design preserves true 1-NFE sampling, followed by a light residual corrector. Appendix 6.2 quantifies Stage-II compute: the default configuration completes under $\sim 900$ seconds with only $\sim 1.15$ GB of memory.
>
> (4) Notation and minor presentation issues.
> We corrected the sign error in Eqs. (12) and (14) and cleaned notation inconsistencies in the revised version.

---

### Official Review · Reviewer_AiFm · 2025-11-03

**Soundness:** 2
**Presentation:** 1
**Contribution:** 1
**Rating:** 2
**Confidence:** 3

**Summary:**

The authors propose a 2-stage algorithm for learning a transport map to solve 3D Keller-Segel problem across different fluid flows. Stage 1 trains a MeanFlow algorithm as one-step transport (1-NFE) that moves prior samples close to the target states. Stage 2 fits a deep particle corrector by directly minimizing a mini-batch $W_2$ objective with warm-started OT couplings, yielding local, geometry-aware refinements.

**Strengths:**

* **Motivation**: Authors propose an interesting problem of optimizing $W_2$ distance between end-point target distribution and MeanFlow predictions in higher dimensions where MeanFlow fails due to not directly targeting $W_2$ distance
* **Clarity**: Authors provide technical background on MeanFlow making the paper accessible to wider audience
* **Results Interpretability**: Results are supported by visualizations making it easy for the reader to understand objectives

**Weaknesses:**

* **Comparison to baselines**: Work lacks comparison to other one-step model baselines. The method is centered around MeanFlow, however it seems that the key novelty is adding deep particle corrector to minimize $W_2$, which is not tied to mean flow as a first stage algorithm. It would be interesting to see what outperformance the method achieves with various one-step models (e.g. consistency models, flow maps via self-distillation [1] or IMM [2]) as stage 1 algorithms.
* **Limited Novelty**: Authors cite [3] in line 229 which if I understand correctly also uses one-step generative models and introduces DP method as a $W_2$ corrector, testing empirically on the similar set of Keller-Segel problems. Given the strong similarities between the two works, they should be further compared in the main text, highlighting key differences and ideally comparing them empirically.
* **Limited Range of Real-world Experiments**: The method is applied across limited range of practical examples. I would encourage authors to discuss in the main text or appendix whether the proposed method could be extended to tackle other high-dimensional problems.

**Questions:**

* In equations (12) and (14), I believe there is a typo? They should read as $x_0 = x_1 - \int_0^1{v(z_t,t)dt}$ (equation 12) and $x_0 = x_1 - u(x_1, 0, 1)$ (equation 13). I believe the same applies to the expression in line 194.
* Do you model higher dimensions ($d>3$)? The work mentions learning in high-dimension setting as one of the core limitation of MeanFlow, and one of the key advantages of DP to correct for $W_2$. It would be useful to see how the method generalizes in higher dimensions and if there are trade-offs between using proposed method and only MeanFlow in terms of accuracy and computational performance. I would suggest authors test ideally in real-world or at least a synthetic setting, and compare their results to MeanFlow.

**Minor comments**
* Figure 1 should perhaps be moved closer to page 1 where it is mentioned

**References**:

[1] Boffi, Nicholas M., Michael S. Albergo, and Eric Vanden-Eijnden. "How to build a consistency model: Learning flow maps via self-distillation." arXiv preprint arXiv:2505.18825 (2025).

[2] Zhou, Linqi, Stefano Ermon, and Jiaming Song. "Inductive moment matching." arXiv preprint arXiv:2503.07565 (2025).

[3] Zhang, Tan, et al. "A Bidirectional DeepParticle Method for Efficiently Solving Low-dimensional Transport Map Problems." arXiv preprint arXiv:2504.11851 (2025).

---

> ### Author Response · Authors · 2025-11-22
>
> Thank you for your thoughtful review and for highlighting the questions about baselines, novelty, and higher-dimensional generalization. Your comments are extremely helpful for improving the clarity and completeness of the paper.
> First, we have fixed the notational/typo issues you pointed out around Eqs. (12) and (14), including the expression for $x_0$ in terms of the velocity field and the definition of $z_t$. We also took this opportunity to polish the notation and references throughout the manuscript.
> We are currently running additional experiments and preparing a revised version of the paper, which we plan to share by November 27. This revision will include:
>
> 1. OT solver ablation: Systematic comparison of EMD vs. regularized Sinkhorn, as well as ablations over coupling refresh frequency and particle numbers.
>
> 2. Stage II scaling curves: A detailed study of runtime, memory consumption, and OT iterations as functions of batch size N.
>
> 3. High-dimensional generalization: New experiments on higher-dimensional synthetic tasks to explicitly demonstrate how the method behaves when d>3.
>
> 4. BDP / MF self-distillation / IMM-style baseline: Some additional one-step generative baselines will be compared directly to our MeanFlow+DP corrector.
>
> 5. Additional σ experiments: Results for other values of the physical parameter σ will be reported in the appendix.
>
> We believe these additions will significantly strengthen the empirical comparison to other one-step models and better position our contribution relative to prior work. Thank you again for the constructive suggestions.

---

> ### Author Response · Authors · 2025-12-03
>
> We thank the reviewer for thoughtful comments on novelty, baselines, and generalization. We have substantially expanded the manuscript and appendix to address these points.
>
> (1) Novelty of the MeanFlow → DP decomposition.
> Our main contribution is a decoupled two-stage design:
> Stage I performs global, deterministic, 1-NFE transport without any OT objective, while Stage II performs a local mini-batch $W_2$ refinement on the already well-aligned geometry. Existing DP-style works directly transport from a Gaussian to data using OT; in contrast, our corrector refines an already-structured parametric map. This turns high-dimensional $W_2$ minimization into a tractable local problem and preserves 1-NFE inference.
>
> (2) Comparison to other one-step baselines.
> Following the reviewer’s suggestion, Appendix 6.3 now includes:
>
> an IMM/self-distillation baseline, adapted to KS with identical architecture and conditioning, and
>
> an SF2M-OT baseline, a state-free flow-matching model trained with Brownian-bridge supervision and evaluated via a 100-step ODE solver.
>
> In all regimes, especially $\sigma \ge 150$, MF+DP achieves the lowest $W_2$, outperforming both IMM and SF2M-OT while retaining single-step sampling.
>
> (3) Higher-dimensional and more diverse experiments.
> To address the concern about $d > 3$ and broader applications, we added:
>
> 4D Keller–Segel (Appendix 6.1), demonstrating that the MF+DP advantage grows with dimension, and
>
> KPP (2D and 3D) experiments (Appendix 6.5), which confirm robust behavior across parameter grids, including extrapolation.
>
> These support the claim that our method generalizes well beyond the original 3D examples.

---

### Official Review · Reviewer_pm4m · 2025-11-04

**Soundness:** 3
**Presentation:** 3
**Contribution:** 3
**Rating:** 6
**Confidence:** 3

**Summary:**

The paper proposes a two-stage, fast-then-precise transport scheme: a MeanFlow initializer deterministically moves particles near the target support, then a near-identity Deep-Particle corrector is trained with mini-batch W₂ using warm-started optimal-transport couplings—preserving one-step sampling while restoring geometry-aware accuracy. After Stage I concentrates mass, the OT plan becomes nearly permutation-like, making W₂ optimization numerically stable and GPU-efficient. The second stage consistently reduces W₂—including under out-of-distribution conditions in stiff Keller–Segel settings—and improves anisotropy and mass placement; a KPP front-speed check suggests these gains translate to downstream physics.

**Strengths:**

On 3D Keller–Segel with flow, DP consistently lowers W₂ across σ and shines in the singular-perturbation/OOD regime (e.g., σ=160: 0.0403→0.0082, σ=200: 0.1970→0.0214). The W₂-vs-σ curve (Figure 2, p. 7) flattens after DP, and projections (Figures 3–4) show better anisotropy and mass placement. A KPP front-speed experiment reveals faster estimator convergence when warm-started by MF→DP, accompanied by a significant drop in W₂ (0.2548→0.02933), indicating that the refinement moves distributions closer in a manner that matters for downstream physics. (Appendix, p. 12.)

**Weaknesses:**

The W₂ refinement still scales quadratically in batch size and depends on solving (mini-batch) OT subproblems. Even after MeanFlow brings supports closer, Stage II must build an O(N²) cost matrix and repeatedly update couplings; the authors explicitly note W₂ is expensive and only “well-suited…in low dimensions,” so scalability in large-N or higher-d regimes remains the bottleneck.

**Questions:**

What are the time/memory costs vs batch size N for Stage II (cost-matrix O(N²), coupling refresh cadence S_γ), and how does training wall-clock compare to a stronger single-step baseline trained longer?

Which OT solver/hyper-params are used (EMD vs Regularized Sinkhorn/interior-point), how often are couplings re-solved during training, and how sensitive are results to entropic regularization or early stopping of the OT subproblem?

When MeanFlow’s initializer is not close (e.g., multimodal supports with large separation), does W₂ training destabilize? Any diagnostics to detect when Stage II should be skipped or down-weighted?

Beyond σ, how does performance vary with particle count, noise in endpoints, and kernel regularization δ in the KS simulator?

---

> ### Author Response · Authors · 2025-11-22
>
> Thank you very much for your careful reading of our paper and for the constructive comments about the OT refinement stage, computational costs, and diagnostics. We really appreciate the detailed questions you raised.
> We are preparing a revised version of the manuscript to be ready by November 27, in which we plan to include the following additional experiments that directly address your concerns:
> 1. OT solver ablation:
> 	Comparison between EMD and entropy-regularized Sinkhorn;
> 	Ablations on coupling refresh frequency and particle numbers to show robustness.
> 2. Stage II scaling curves:
> 	Detailed runtime / memory / number of OT iterations vs. batch size N.
> 3. High-dimensional generalization:
> 	Experiments on higher-dimensional synthetic problems to illustrate how the method behaves for d>3.
> 4. Additional σ settings:
> 	We will report simulation results for several other values of σin the appendix, to show how performance depends on the physical parameter.
>
> We hope these additions will clarify the computational behavior of Stage II and answer your questions about robustness and scalability. Thank you again for the very helpful feedback.

---

> ### Author Response · Authors · 2025-12-03
>
> We thank the reviewer for the detailed and constructive comments. Below we address each point and summarize the substantial new experiments added to the appendix.
>
> (1) Time/memory cost of Stage II (Wasserstein refinement).
> To address the concern about the $O(N^2)$ cost in Stage II, we added an extensive ablation study in Appendix 6.2. We sweep the DP mini-batch size $B_{\mathrm{dp}} \in {1500, 2000, \dots, 3500}$ and the coupling-refresh period $\gamma_{\mathrm{renew}} \in {5, 25, 50, 100}$.
>
> Across all $\sigma$, changing $B_{\mathrm{dp}}$ alters the final $W_2$ only at the $10^{-3}$ level, while the wall-clock cost scales nearly linearly. The configuration $(B_{\mathrm{dp}} = 1500, \gamma_{\mathrm{renew}} = 50)$ achieves nearly optimal $W_2$ while keeping Stage II training under $\sim 900$ seconds and peak GPU memory at $\approx 1.15$ GB. This supports our claim that Stage II remains lightweight, especially relative to full OT training.
>
> (2) OT solver choice and sensitivity.
> Appendix 6.4 compares our exact mini-batch EMD solver with regularized Sinkhorn ($\varepsilon \in {0.05, 0.01, 0.005, 0.001}$). Large $\varepsilon$ over-smooths the transport plan and increases $W_2$, whereas very small $\varepsilon$ approximates EMD but becomes more than $10\times$ slower and uses more memory. Because MeanFlow already moves distributions close to one another, the OT geometry becomes nearly permutation-like, and exact EMD becomes both more accurate and more efficient. We therefore adopt EMD as the default solver.
>
> (3) Stability when MeanFlow is imperfect; generalization across parameters and dimension.
> We added three new experiments:
>
> Harder 3D regimes. In the singular-perturbation range ($\sigma \ge 150$), MeanFlow’s error rises sharply, whereas MF+DP remains flat and low (e.g., $0.0403 \rightarrow 0.0082$ at $\sigma = 160$).
>
> 4D Keller–Segel. Appendix 6.1 extends our method to $d = 4$. MeanFlow’s $W_2$ explodes in the stiff regime (e.g., $0.1526$ at $\sigma = 200$), while DP stays nearly constant ($0.0134$). This demonstrates that our two-stage design is even more beneficial as dimension increases.
>
> KPP front-speed experiments (2D & 3D). Appendix 6.5 evaluates downstream physics. MF+DP produces invariant measures significantly closer to the long-horizon Feynman–Kac reference and yields faster eigenvalue convergence, even at extrapolation diffusion levels.
>
> These results show that Stage II remains stable and useful exactly when Stage I has residual bias, and they highlight strong generalization across $\sigma$, flows, and dimension while keeping inference 1-NFE.

---

### Author Response · Authors · 2025-12-03

We thank the reviewers for their careful and constructive feedback. Below we summarize the major revisions and how they address the stated concerns.

**Novelty clarification.**
We clarified that our contribution is a *decoupled, two-stage coarse-to-fine pipeline*: MeanFlow provides fast 1-NFE global transport, while a near-identity DeepParticle corrector performs local mini-batch $W_2$ refinement. This design is fundamentally different from prior one-step regression approaches.

**New experiments added to address generalization.**

- **4D Keller–Segel (Appendix 6.1).**
  MeanFlow errors increase rapidly in stiff high-σ regimes (e.g., 0.1526 at σ=200), whereas DP refinement stays nearly flat (~0.0134). This demonstrates favorable scaling to higher dimensions.

- **2D/3D KPP front-speed (Appendix 6.5).**
  Using our generator as a warm start yields substantially improved invariant-measure accuracy and eigenvalue convergence, including at diffusion levels outside the training range.

These results collectively show strong generalization across PDE types, dimensions, and physical parameters, while preserving one-shot (1-NFE) sampling.

**Additional baselines included.**

- IMM / self-distilled flow map
- SF2M-OT (Brownian-bridge flow matching)

Across all σ, including stiff extrapolated regimes, MF+DP consistently achieves the lowest $W_2$ while maintaining deterministic, single-step inference.

**Ablation studies added to address computational concerns.**

- **Batch size & coupling refresh (Appendix 6.2).**
  The default configuration (B_dp=1500, γ_renew=50) achieves near-optimal $W_2$ with <900s runtime and ~1.15 GB GPU memory.

- **Sinkhorn vs. EMD (Appendix 6.4).**
  After MeanFlow alignment, EMD is both faster and more accurate than all regularized Sinkhorn variants. Small-ε Sinkhorn becomes prohibitively slow.

**Other improvements.**
We corrected typos in Eqs. (12) and (14), improved clarity in several sections, and reorganized figures for easier comparison.

Overall, we believe the revision fully addresses all reviewer concerns and significantly strengthens the submission.

---

### Meta-Review · Area_Chair_HtXf · 2026-01-05

**Summary:**

The paper proposes a two-stage process to accurately map an initial set of particles into terminal stage within the context of fluid flow phenomena. The proposed method outperforms prior work MeanFlow in matching the target distribution under range of physical parameters. This outperformance is captured using the Wasserstein distance with respect to a target distribution.  The proposed methods consistency outperforms the prior work on numerical experiments including the 3D and 4D Keller Segel Systems and Kolmogorov-Petrovksy-Piskunov front speed experiment.

First, I would like to acknowledge the work that the authors put to demonstrate the efficacy and scaling capabilities of their proposed method. However, it is not very clear what contributions from this work could be of interest to the general ML community.  Moreover, the motivation is not very clear, or it is not discussed in a convincing way. The paper jumps directly into the problem giving the impression that the reliable prediction of 3D Keller-SeGel Chemotaxis systems is a well-known and broadly understood problem.  Why is this an important problem to solve for fluid flows and what problems other than fluid flows could one address with the proposed practical algorithm? Given the lack of some theoretical analysis, adding results on experiments other than fluid flow such as image generation could have further improved the paper. The MeanFlow cited by the authors work includes such experiments. Therefore, it would make sense that the authors add this use case to their comparisons to strengthen the broad impact of their proposed algorithm.  For the reasons above I recommend rejection.

Also, in the paper I do not see ethics and reproducibility statement.

**Reviewer Concerns:**

Most of the reviewer concerns were addressed by the authors however, I would like to point out that the applications in the paper are very specific and cover a  narrow area of  potential use cases of the proposed method. This makes the reviewers not confident in their assessment in terms of how important the problem in consideration is. The fundamental issue with this paper is pointed by reviewer g1Pi and I actually I agree with this point. As pointed out by the reviewer:

" 3D Keller-Segel chemotaxis is a very specific application. I think this framework could generalize to different problems (e.g. simulation-based inference, cell dynamics, etc.). A wider focus on the experiment/application side would make this paper more approachable for the community. It is also difficult for me to assess the experiment setup and scientific importance, since I am not familiar with Keller-Segel Chemotaxis."

In addition the point raised by reviewer AiFm regarding novelty is still valid.

The aforementioned two concerns remain outstanding.

**Reviewer Scores:**

I am not certain that the reviewers would have changed the score.  I believe that the reviewers would have maintain their scores.

---

### Decision · Program_Chairs · 2026-01-26

Reject